# A molecular pathology, neurobiology, biochemical, genetic and neuroimaging study of progressive apraxia of speech

Keith A. Josephs [1✉], Joseph R. Duffy[1], Heather M. Clark[1], Rene L. Utianski[1], Edythe A. Strand[1,2], Mary M. Machulda[3], Hugo Botha [1], Peter R. Martin [4], Nha Trang Thu Pham[5], Julie Stierwalt[1], Farwa Ali[1], Marina Buciuc[1], Matthew Baker[6], Cristhoper H. Fernandez De Castro[6], Anthony J. Spychalla[5], Christopher G. Schwarz [5], Robert I. Reid [7], Matthew L. Senjem[5,7], Clifford R. Jack Jr [5], Val J. Lowe[5], Eileen H. Bigio[8], Ross R. Reichard[9], Eric. J. Polley[4], Nilufer Ertekin-Taner[6], Rosa Rademakers[6,10], Michael A. DeTure[6], Owen A. Ross [6], Dennis W. Dickson [6] & Jennifer L. Whitwell[5]

Progressive apraxia of speech is a neurodegenerative syndrome affecting spoken communication. Molecular pathology, biochemistry, genetics, and longitudinal imaging were investigated in 32 autopsy-confirmed patients with progressive apraxia of speech who were followed over 10 years. Corticobasal degeneration and progressive supranuclear palsy (4R-tauopathies) were the most common underlying pathologies. Perceptually distinct speech characteristics, combined with age-at-onset, predicted specific 4R-tauopathy; phonetic subtype and younger age predicted corticobasal degeneration, and prosodic subtype and older age predicted progressive supranuclear palsy. Phonetic and prosodic subtypes showed differing relationships within the cortico-striato-pallido-nigro-luysial network. Biochemical analysis revealed no distinct differences in aggregated 4R-tau while tau H1 haplotype frequency (69%) was lower compared to 1000+ autopsy-confirmed 4R-tauopathies. Corticobasal degeneration patients had faster rates of decline, greater cortical degeneration, and shorter illness duration than progressive supranuclear palsy. These findings help define the pathobiology of progressive apraxia of speech and may have consequences for development of 4R-tau targeting treatment.

[1] Department of Neurology, Mayo Clinic, Rochester, MN, USA. [2] Speech & Hearing Sciences, University of Washington, Seattle, WA, USA. [3] Department of Psychiatry and Psychology, Mayo Clinic, Rochester, MN, USA. [4] Department of Quantitative Health Sciences (Biostatistics), Mayo Clinic, Rochester, MN, USA. [5] Department of Radiology, Mayo Clinic, Rochester, MN, USA. [6] Department of Neuroscience, Jacksonville, FL, USA. [7] Department of Information Technology, Rochester, MN, USA. [8] Department of Pathology, Northwestern University School of Medicine, Chicago, IL, USA. [9] Department of Laboratory Medicine and Pathology, Mayo Clinic, Rochester, MN, USA. [10] VIB-UA Center for Molecular Neurology, VIB, University of Antwerp, Antwerp, Belgium. ✉email: josephs.keith@mayo.edu

In 1967, Fred Darley described a motor speech disorder characterized by varying combinations of slow speaking rate; abnormal prosody; speech sound simplifications; distorted sound substitutions, additions, repetitions, and prolongations; segmentations between syllables and words; and groping and trial-and-error articulatory movements and coined the term apraxia of speech (AOS)[1]. AOS was designated as a unique entity distinct from dysarthria (another type of motor speech disorder) and aphasia, presumably due to abnormal planning and/or programming of speech movements[2]. Patients typically exhibit frustration with their inability to correct errors and often resort to more carefully articulated speech and/or slowing of their speech rate.

Stroke-related AOS, which typically improves over time, is well recognized in the literature[2,3]. However, in the past two decades, a subtype of AOS has been described that is insidious in onset and progressive in nature[4–6]. This progressive AOS (PAOS) often accompanies progressive agrammatic aphasia[7] (AOS-PAA) or occurs as an embedded feature of a more widespread neurodegenerative syndrome, such as corticobasal syndrome (CBS) or amyotrophic lateral sclerosis[8] (referred to here as +AOS). In 2005, the term primary progressive apraxia of speech (PPAOS) was coined by Duffy et al. to describe PAOS occurring in its relatively pure form, i.e., in the absence of aphasia and not embedded as a feature of a larger neurodegenerative syndrome[5]. The syndrome of PPAOS was later more fully characterized and shown to relate to degeneration of a network of regions that include the lateral premotor cortex and supplementary motor area (SMA)[9]. The prevalence of PAOS, which encompasses PPAOS, has been estimated to be about 4.4 per 100,000 individuals [10].

In 2013, Josephs et al. described two data-supported subtypes of PAOS[11]. The first, type 1, was characterized by the relative predominance of articulatory distortions, distorted sound substitutions or additions, and articulatory groping. This subtype is similar to what has been reported in patients with stroke. The second, type 2, was characterized by a predominance of slow speech rate and segmentation within multi-syllabic words or across words. Type 2 has not been described in stroke patients and was not previously formally recognized. In subsequent studies, type 1 has been referred to as the phonetic subtype, and type 2 as the prosodic subtype[12]. Features supporting the validity of PAOS subtypes include the phonetic subtype being associated with a significantly younger age at onset and more likely accompanied by moderate-severe aphasia than the prosodic subtype[11,12]. In fact, the literature describing PAOS accompanying aphasia, or AOS embedded in the context of a neurodegenerative syndrome, almost always reveals characteristics consistent with phonetic AOS. Both PAOS subtypes involve degeneration of the lateral premotor and SMA, although the phonetic subtype appears to be more strongly linked to neocortex while the prosodic subtype has been linked to noncortical regions, such as the superior cerebellar peduncle [12].

The motor speech impairment in PAOS progressively worsens over time, eventually resulting in mutism. In addition, patients with PAOS, including those with PPAOS, typically evolve into a Parkinson-plus syndrome[13]. Parkinsonian features tend to become prominent around 5 years after onset and include bradykinesia, postural instability, and rigidity, but not tremor[14–16]. Patients with PAOS may also develop ideomotor limb apraxia[13–15] and nonverbal oral apraxia (orobuccal apraxia)[17]; a subgroup develops ocular motor abnormalities including oculomotor impersistence and vertical supranuclear gaze palsy[13,14,16]. Interestingly, the development of parkinsonism has been particularly linked to the prosodic subtype of PAOS[18]. In keeping with the clinical progression is the presence of neuroanatomic progression where progressive degeneration is evident in premotor and motor cortices as well as striatum, globus pallidus, and midbrain, particularly in those patients who develop a Parkinson-plus syndrome [13,14,18].

In a handful of retrospective case reports and small case series, the underlying pathology associated with PAOS was a tauopathy, particularly a 4-repeat (4R) tauopathy, such as progressive supranuclear palsy (PSP) or corticobasal degeneration (CBD)[4,6,19]. It is unknown, however, whether molecular pathology differs when AOS: (1) presents as a pure entity (PPAOS); (2) is associated with aphasia; or (3) is embedded within another neurodegenerative syndrome. It is also unknown whether there is any relationship between molecular pathology and AOS subtype or whether other clinical or neuroimaging features help predict underlying pathology. Furthermore, there is a knowledge gap concerning whether genetic associations, such as the H1/H2 haplotype of the microtubule associated tau (MAPT) gene[20,21] or the transmembrane protein 106B (TMEM106B), and biochemical features of aggregated tau associated with PSP and CBD pathology[22] are associated with 4R tauopathies underlying PAOS, or whether genetic mutations or polymorphisms in TMEM106B associated with degenerative diseases such as PSP, CBD, Picks disease (PiD) and the protein TDP-43, play any role in PAOS.

In this study, we report the molecular pathology of a relatively large cohort of 32 prospectively recruited and longitudinally followed patients with PAOS. We aimed to investigate clinical, neuroimaging, genetic, and biochemical associations with molecular pathology in PAOS. We hypothesized that PSP and CBD would be the most common underlying pathology of PPAOS and PAOS with aphasia. Given that we found an association between phonetic PAOS and anatomical involvement of the premotor cortex and involvement of the superior cerebellar peduncle in prosodic AOS[12], we hypothesize that phonetic AOS would be associated with CBD while prosodic AOS would be associated with PSP and that neuroimaging features will differ between CBD and PSP, especially closer to death. We hypothesize that genetic and biochemical characteristics of PAOS-CBD and PAOS-PSP will be similar to those previously reported in CBD and PSP when associated with their more classic presentations of corticobasal syndrome (CBS) and Richardson's syndrome, respectively.

## Results

**Cohort characteristics**. Our cohort was composed of 16 women and 16 men. All but five were right-handed, and all were White, non-Hispanic. Median education was 15.5 years (range 12–20), and age at death was 71 years (range 46–86).

**Pathological diagnoses**. Of the 32 PAOS patients, the most common underlying pathology was CBD, which was found in 17 cases (53%), followed by PSP, which was observed in 10 cases (31%, Fig. 1). All CBD and PSP patients met research criteria[23,24]. Less frequent pathologies were frontotemporal lobar degeneration with TDP-43 (FTLD-TDP) type A in two cases (6%), PiD in two cases (6%), and combined FTLD-TDP type B with argyrophilic grains disease (AGD) in one case (3%).

Of the ten patients with PSP, three met pathologic criteria for atypical PSP[24]. All three showed cortical predominant pathology with relatively mild neuronal loss in cardinal subcortical nuclei (i.e., globus pallidus, subthalamic nucleus and substantia nigra). Two of the three atypical PSP patients additionally had a loss of Betz cells in the motor cortex, with one additionally having globose neurofibrillary tangles (NFTs) within residual Betz cells that were apparent on H&E and with thioflavin S fluorescent microscopy (Supplementary Fig. 1). In addition, this patient showed many tau-positive threads in the posterior limb of the

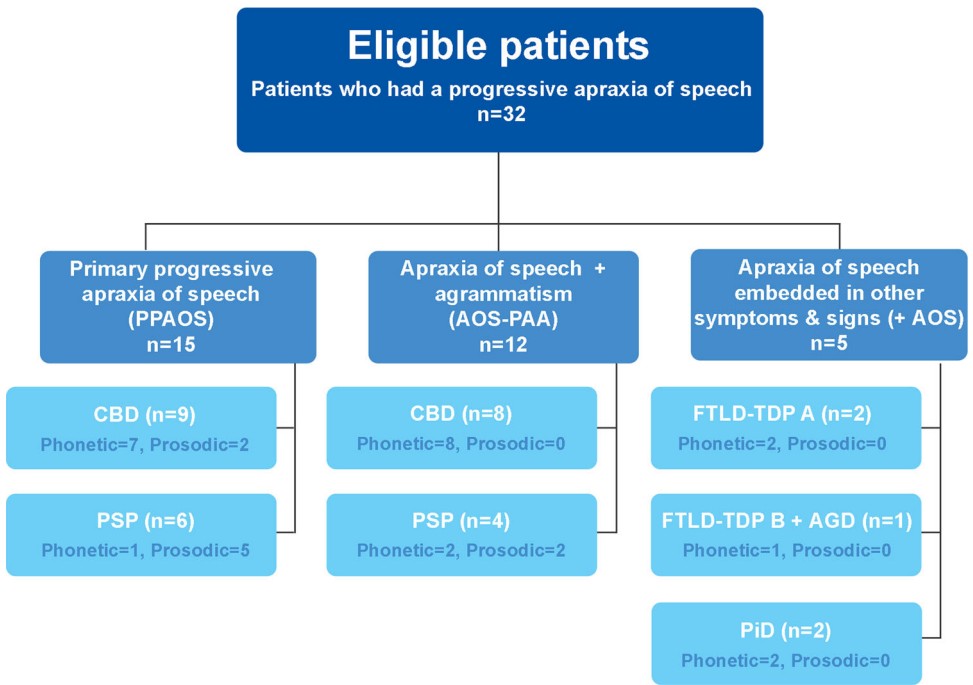

**Fig. 1 Flowchart illustrating the relationship between baseline clinical diagnoses and underlying pathology in all 32 progressive apraxia of speech patients.** Thirty-two patients with progressive apraxia of speech were included in the study. Fifteen of these patients were diagnosed with primary progressive apraxia of speech (PPAOS), of which nine had corticobasal degeneration (CBD) pathology and six had progressive supranuclear palsy (PSP) pathology. Twelve of the 32 patients were diagnosed with both apraxia of speech and agrammatism (AOS-PAA), of which 8 had CBD and 4 had PSP pathology. The last five patients were diagnosed with apraxia of speech that was embedded in other symptoms and signs (+AOS). These five patients had frontotemporal lobar degeneration pathology with TAR DNA binding of 43Kda (FTLD-TDP) (either type A or B) and Pick disease (PiD) pathologies, with one patient also having argyrophilic grains disease (AGD). The number of patients with the phonetic and prosodic subtype of apraxia of speech is shown in each box.

internal capsule, mid-third of the cerebral peduncle, and the longitudinal fibers in the pontine base but less in the medullary pyramid, consistent with atypical PSP with corticospinal tract degeneration[25]. None of the PSP patients had globular inclusions consistent with a globular glial tauopathy[26]. Of the 17 patients with CBD, the pathology was typical in all but two patients, who had additional features. One had striking neuronal loss and gliosis in the globus pallidus, substantia nigra, and subthalamic nucleus, accompanied by iron-type pigment deposits in the globus pallidus, consistent with pallidonigraluysian degeneration (PNLD)[27,28]. The second patient showed striking neuronal loss, gliosis, iron-type pigment, and axonal spheroids in the globus pallidus, plus marked neuronal loss in the substantia nigra but less pathology in the subthalamic nucleus.

No case met criteria for high likelihood of Alzheimer's disease neuropathologic changes[29] (Supplementary Table 1).

**Demographic and clinical features of PAOS-CBD and PAOS-PSP at baseline.** The PAOS-CBD and PAOS-PSP cases all had a clinical diagnosis of either PPAOS or AOS-PAA at their baseline visit, while the FTLD-TDP type A, FTLD-TDP type B + AGD, and PiD cases all had a diagnosis of +AOS at baseline (Fig. 1). Patients diagnosed with PPAOS and AOS-PAA at baseline showed a similar proportion of CBD and PSP pathologies at autopsy (60% CBD and 40% PSP in PPAOS; 67% CBD and 33% PSP in AOS-PAA).

The PAOS-PSP cases were older at onset, baseline evaluation, and death, and they had a longer time from onset to death compared to PAOS-CBD cases (Table 1). The AOS subtype differed across PAOS-PSP and PAOS-CBD; the phonetic subtype was more common in PAOS-CBD, and the prosodic subtype was

more common in PAOS-PSP. In fact, of the 17 PAOS-CBD cases, 15 (88%) had phonetic AOS, while of the 10 PSP cases only 3 (30%) had phonetic AOS. The three PAOS-PSP patients with phonetic AOS were all diagnosed with atypical PSP at autopsy (with severe cortical pathology and less severe subcortical pathology, as described above). The other seven had typical PSP pathology. The two PAOS-CBD patients with prosodic AOS showed additional features (including striking involvement of the globus pallidus, substantia nigra, and subthalamic nucleus, as described above). The PAOS-CBD group also performed worse on the phonetic subscore of the AOS rating scale (ASRS) compared to PAOS-PSP cases. A logistic regression model using elastic-net regularization predicting pathology using age at onset and AOS subtype gave an AUROC of 0.89 (0.68, 0.97). Supplementary Table 2 provides the probabilities that a PAOS patient has either CBD or PSP pathology given their age at onset and AOS subtype. For example, a patient with an age of onset of 60 years and phonetic AOS has an 84% probability of having CBD pathology. Conversely, a patient with an age at onset of 80 years and prosodic AOS has a 93% probability of having PSP pathology. The only clinical tests that differed across PAOS-CBD and PAOS-PSP were the Frontal Behavioral Inventory (FBI), Boston Naming Test (BNT), and Delis–Kaplan Executive Function System (DKEFS) Sorting Test, on which the PAOS-CBD cases performed worse.

**Longitudinal clinical progression in PAOS-CBD and PAOS-PSP.** The PAOS-CBD patients showed greater rates of worsening of aphasia, motor speech, confrontation naming, general cognition, and visual memory, compared to the PAOS-PSP patients (Table 1). Over time, of the 17 PAOS-CBD patients, 13 evolved

**Table 1 Baseline demographic and clinical features for the PAOS-CBD and PAOS-PSP cases.**

| | Baseline data | | | | Annualized rates of change | | |
|---|---|---|---|---|---|---|---|
| | PAOS-CBD (N = 17) | PAOS-PSP (N = 10) | p value | AUROC (95% CI) | PAOS-CBD (N = 12) | PAOS-PSP (N = 9) | p value |
| **Demographics** | | | | | | | |
| Female sex, N % | 8 (47%) | 4 (40%) | >0.999 | 0.54 (0.32/0.73) | NA | NA | NA |
| Education, years | 16 (12, 16) | 16 (15, 20) | 0.165 | 0.66 (0.43/0.83) | NA | NA | NA |
| Handedness (L/R/A), N | 2/15/0 | 2/7/1 | 0.385 | 0.59 (0.37/0.77) | NA | NA | NA |
| Age onset, years | 60 (51, 64) | 72 (65, 74) | 0.003 | 0.85 (0.63/0.95) | NA | NA | NA |
| Age death, years | 69 (63, 73) | 83 (75, 84) | 0.001 | 0.88 (0.67/0.96) | NA | NA | NA |
| Duration (Onset to death), years | 9 (7, 10) | 11 (10, 13) | 0.021 | 0.77 (0.54/0.90) | NA | NA | NA |
| Age at baseline, years | 64 (60, 69) | 75 (71, 78) | 0.002 | 0.87 (0.65/0.96) | NA | NA | NA |
| Onset to baseline, years | 5 (3, 6) | 5 (3, 6) | 0.960 | 0.51 (0.30/0.71) | NA | NA | NA |
| Baseline to last evaluation, years | NA | NA | NA | NA | 2.7 (1.8, 3.5) | 5.3 (3.5, 5.7) | 0.009 |
| Number of serial visits | NA | NA | NA | NA | 2 (1, 4) | 5 (3, 6) | NA |
| Number of serial MRI | NA | NA | NA | NA | 2 (1, 4) | 4 (3, 5) | NA |
| Number of serial PET | NA | NA | NA | NA | 2 (1, 3) | 4 (2, 5) | NA |
| **Genetic data** | | | | | | | |
| Family history in 1st or 2nd degree relative, N % | 1 (6%) | 2 (20%) | 0.535 | 0.57 (0.35/0.76) | NA | NA | NA |
| TMEM106B CC/CG/GG | 6/10/1 | 2/7/1 | 0.836 | 0.58 (0.36/0.77) | NA | NA | NA |
| GRN/MAPT/C9ORF72, N | 0/0/0 | 0/0/0 | NA | NA | NA | NA | NA |
| APOE ε4, N % | 4 (24%) | 1 (10%) | 0.621 | 0.57 (0.35/0.76) | NA | NA | NA |
| H1H1 haplotype, N % | 15 (88%) | 4 (40%) | 0.024 | 0.74 (0.51/0.88) | NA | NA | NA |
| **Speech and language data** | | | | | | | |
| Phonetic/prosodic AOS type, N | 15/2 | 3/7 | 0.004 | 0.79 (0.56/0.91) | NA | NA | NA |
| Aphasia present, N % | 9 (53%) | 4 (40%) | 0.695 | 0.56 (0.35/0.76) | NA | NA | NA |
| Aphasia severity/4 | 0 (0, 1) | 0 (0, 1) | 0.851 | 0.52 (0.31/0.72) | 0.6 (0.4, 0.6) | 0.3 (0.2, 0.4) | 0.006 |
| WAB Aphasia Quotient, /100 | 95 (91, 96) | 97 (87, 100) | 0.225 | 0.64 (0.42/0.82) | −6.7 (−13.4, −5.5) | −1.4 (−2.9, −1.1) | <0.001 |
| ASRS Total Score V3 | 19 (14, 26) | 16 (11, 22) | 0.526 | 0.57 (0.35/0.77) | 4.5 (2.0, 4.9) | 4.2 (3.0, 4.4) | 0.691 |
| ASRS Phonetic subscore | 6 (5, 11) | 4 (3, 4) | 0.013 | 0.79 (0.56/0.91) | 1.5 (0.0, 2.5) | 1.4 (0.9, 1.7) | 0.965 |
| ASRS Prosodic subscore | 6 (4, 8) | 8 (3, 10) | 0.916 | 0.51 (0.30/0.72) | 1.1 (0.9, 1.2) | 0.8 (0.6, 2.2) | 0.895 |
| MSD Severity Scale, /10 | 6 (5, 7) | 7 (6, 8) | 0.085 | 0.70 (0.47/0.85) | −1.2 (−1.5, −1.0) | −0.6 (−0.9, −0.5) | 0.004 |
| BNT, /15 | 13 (12, 14) | 15 (14, 15) | 0.010 | 0.80 (0.57/0.92) | −1.2 (−2.3, −0.9) | −0.3 (−0.7, −0.1) | 0.027 |
| WAB fluency, /10 | 9 (8, 10) | 9 (9, 10) | 0.779 | 0.53 (0.32/0.73) | −1.5 (−2.0, −1.0) | −0.5 (−0.9, −0.1) | 0.025 |
| Letter fluency (FAS) | 17 (10, 25) | 18 (18, 27) | 0.413 | 0.60 (0.37/0.78) | −4.2 (−4.9, −0.6) | −2.9 (−5.0, −1.6) | 0.958 |
| WAB animal fluency | 14 (11, 15) | 16 (11, 20) | 0.278 | 0.63 (0.40/0.81) | −3.0 (−4.0, −2.5) | −2.4 (−2.8, −1.6) | 0.070 |
| Token Test, /22 | 20 (12, 21) | 20 (19, 22) | 0.129 | 0.69 (0.44/0.86) | −1.7 (−3.0, −0.7) | −1.7 (−3.0, −0.3) | 0.501 |
| NAT, /10 | 8 (6, 10) | 9 (9, 10) | 0.413 | 0.61 (0.35/0.82) | −0.8 (−2.8, −0.0) | −0.5 (−1.0, −0.1) | 0.700 |
| PPT word-word, /52 | 50 (48, 51) | 50 (49, 51) | 0.268 | 0.63 (0.40/0.81) | −0.8 (−1.7, −0.1) | −0.1 (−0.6, 0.0) | 0.138 |
| Dysarthria present, N % | 4 (24%) | 2 (20%) | >0.999 | 0.52 (0.31/0.72) | NA | NA | NA |
| Dysarthria severity, /4 | 0 (0, 0) | 0 (0, 0) | 0.974 | 0.50 (0.30/0.71) | 0.4 (0.2, 0.7) | 0.3 (0.2, 0.4) | 0.676 |
| NVOA, /32, 32 = normal | 22 (15, 30) | 29 (26, 32) | 0.277 | 0.63 (0.40/0.81) | −4.4 (−6.5, −3.5) | −5.6 (−6.5, −5.3) | 0.522 |
| **Neurological/neuropsychological data** | | | | | | | |
| MMSE, /30 | 29 (29, 29) | 30 (28, 30) | 0.345 | 0.61 (0.38/0.79) | −2.9 (−4.8, −1.8) | −0.6 (−0.8, −0.3) | 0.005 |
| MoCA, /30 | 25 (24, 26) | 28 (25, 28) | 0.287 | 0.62 (0.40/0.80) | −3.2 (−5.6, −2.6) | −1.7 (−2.0, −1.2) | 0.011 |
| FAB, /18 | 15 (14, 17) | 16 (14, 17) | 0.503 | 0.58 (0.36/0.77) | −2.3 (−4.3, −1.7) | −1.9 (−2.7, −0.9) | 0.394 |
| FBI, /72, 0 = best | 12 (5, 18) | 4 (2, 5) | 0.001 | 0.88 (0.66/0.96) | 5.9 (3.0, 8.9) | 3.7 (1.8, 5.3) | 0.283 |
| NPI-Q, /36, 0 = best | 2 (1, 6) | 2 (0, 3) | 0.259 | 0.63 (0.40/0.81) | 1.2 (0.3, 4.6) | 0.5 (0.3, 0.8) | 0.186 |

**Table 1 (continued)**

| | Baseline data | | | | Annualized rates of change | | |
| --- | --- | --- | --- | --- | --- | --- | --- |
| | PAOS-CBD (N = 17) | PAOS-PSP (N = 10) | p value | AUROC (95% CI) | PAOS-CBD (N = 12) | PAOS-PSP (N = 9) | p value |
| MDS-UPDRS III, /132, 0 = best | 14 (5, 23) | 14 (8, 22) | >0.999 | 0.50 (0.29/0.71) | 12.2 (7.7, 18.4) | 12.3 (8.0, 15.4) | 0.943 |
| WAB Praxis, /60 | 56 (53, 58) | 58 (52, 59) | 0.730 | 0.54 (0.33/0.74) | −7.0 (−10.7, −5.4) | −5.4 (−6.3, −1.5) | 0.177 |
| PSIS, /5, 0=best | 0 (0, 1) | 1 (0, 1) | 0.154 | 0.65 (0.42/0.82) | 0.4 (0.2, 0.5) | 0.3 (0.2, 0.5) | >0.999 |
| TMT A MOANS | 6 (5, 9) | 8 (6, 10) | 0.380 | 0.60 (0.38/0.79) | −0.6 (−2.6, 0.0) | −1.8 (−1.9, −0.4) | 0.817 |
| TMT B MOANS | 7 (5, 8) | 10 (8, 10) | 0.111 | 0.70 (0.45/0.87) | −1.5 (−2.0, −0.8) | −1.3 (−2.1, −1.0) | 0.668 |
| Sorting Test SS | 9 (8, 12) | 12 (11, 16) | 0.042 | 0.75 (0.51/0.89) | −0.3 (−0.8, −0.0) | −0.7 (−1.3, 0.2) | 0.699 |
| WMS-III VR I SS | 8 (6, 11) | 11 (9, 14) | 0.093 | 0.72 (0.46/0.88) | 0.0 (−0.4, 0.4) | 0.3 (−0.4, 0.5) | 0.643 |
| WMS-III VR II SS | 12 (9, 14) | 14 (12, 15) | 0.149 | 0.69 (0.43/0.86) | −1.7 (−3.9, −1.4) | 0.2 (−0.5, 0.7) | 0.021 |
| VOSP letters, /20 | 20 (19, 20) | 20 (20, 20) | 0.162 | 0.64 (0.42/0.82) | 0.0 (−0.0, 0.3) | −0.0 (−0.2, 0.0) | 0.282 |
| VOSP cubes, /10 | 10 (9, 10) | 9 (8, 10) | 0.349 | 0.60 (0.38/0.79) | 0.0 (−1.9, 0.0) | −0.1 (−0.5, 0.0) | 0.690 |

Data shown as median (inter-quartile range) or N (%). Annualized rates of change are calculated as average change overall visits within participant.
MOANS and SS have a mean of 10 with a standard deviation of 3 in normal healthy individuals.
L left, R right, A ambidextrous, APOE apolipoprotein, GRN progranulin, MAPT microtubule associated protein tau, MMSE Mini Mental State Examination, MoCA Montreal Cognitive Assessment, FAB Frontal Assessment Battery, FBI Frontal Behavioral Inventory, NPI-Q short questionnaire version of the Neuropsychiatric Inventory, MDS-UPDRS III Movement Disorder Society sponsored revision of the Unified Parkinson's Disease Rating Scale part III, WAB Western Aphasia Battery, PSIS Progressive supranuclear palsy Saccadic Impairment Scale, AOS apraxia of speech, ASRS Apraxia of Speech Rating Scale, MSD Motor Speech Disorder, BNT Boston Naming Test, NAT Northwestern Anagram Test, PPT Pyramids and Palm Trees test, NVOA nonverbal oral apraxia, SS scaled score, TMT Trail Making Test, WMS-III VR Wechsler Memory Scale-III Visual Reproduction, MOANS Mayo Older American Normative Studies, VOSP Visual Object and Space Perception Battery.

into a CBS[30], two into behavioral variant frontotemporal dementia[31], one into frontotemporal dementia with parkinsonism (i.e., in addition to features of behavioral variant FTD, he also developed early in his course levodopa-unresponsive parkinsonism of moderate severity), and one into Richardson's syndrome[32]. Of the 10 PAOS-PSP patients, five evolved into Richardson's syndrome and three into a CBS; the final two patients did not have longitudinal follow-up. The AOS subtype progressed to become mixed (i.e. ~equal prosodic and phonetic features) in 15 of 23 patients (65%) with follow-up clinical assessments, without discernable differences in conversion between PAOS-CBD and PAOS-PSP or between phonetic and prosodic patients. Of the remaining eight patients, none had converted to a mixed phenotype by the last clinical follow-up, although all patients, except one with CBS, became mute before death.

Longitudinal trajectories of decline in cognition, parkinsonism, global aphasia, and AOS across patients are shown in Fig. 2, with model estimates provided in Supplementary Table 3. Change over time in the Montreal Cognitive Assessment Battery (MoCA), ASRS and Western Aphasia Battery (WAB) aphasia quotient did not reach significance for the nonlinear model, with evidence for a difference in the rate of decline observed between PAOS-CBD and PAOS-PSP for only the WAB aphasia quotient; PAOS-CBD showed faster rates of decline than PAOS-PSP. Increase in the Movement Disorders Society-sponsored revision of the Unified Parkinson's Disease Rating Scale (MDS-UPDRS III) was nonlinear, with accelerating rates of change over time and no evidence for differences between PAOS-CBD and PAOS-PSP.

**Genetic findings in PAOS-CBD and PAOS-PSP.** None of the PAOS-CBD and PAOS-PSP patients had mutations in progranulin (GRN), MAPT or C9ORF72, and we did not see any difference in TMEM106B genotype or APOE ε4 frequency across PAOS-CBD and PAOS-PSP (Table 1). There was, however, a difference in tau haplotype frequency, with the H1H1 haplotype being more common in PAOS-CBD (88%) compared to PAOS-PSP (40%, Table 1). Two PAOS-CBD and five PAOS-PSP patients were H1H2, and one PAOS-PSP patient was H2H2. The relative frequency of the H1H1 haplotype to the H1H2 haplotype in PAOS-CBD was similar to that observed in the 230 CBS-CBD cases (H1H1 = 87% versus 88% PAOS-CBD). On the other hand, the relative frequency of the H1H1 haplotype in PAOS-PSP was different from that observed in the 802 PSP cases with Richardson's syndrome (H1H1 = 40% PAOS-PSP versus 90%).

**Voxel-level neuroimaging findings at baseline in PAOS-CBD and PAOS-PSP.** PAOS-CBD and PAOS-PSP both showed bilateral hypometabolism in lateral and medial premotor cortex compared to controls, although PAOS-CBD showed more widespread hypometabolism within these regions that survived correction for multiple comparisons and involvement of striatum and thalamus (Fig. 3). No differences were observed between PAOS-CBD and PAOS-PSP on direct comparison.

PAOS-CBD and PAOS-PSP both showed reduced fractional anisotropy (FA) compared to controls (uncorrected $p < 0.001$) in the body of the corpus callosum, cingulum, superior corona radiata, and superior frontal, precentral and postcentral white matter (corrected for multiple comparisons) (Fig. 4). Additional regions that showed reduced FA in PAOS-PSP only compared to controls included midbrain and superior cerebellar peduncle (corrected for multiple comparisons). Similar but slightly less widespread findings were observed with mean diffusivity (MD). No differences in FA or MD were identified between PAOS-CBD and PAOS-PSP on direct comparison.

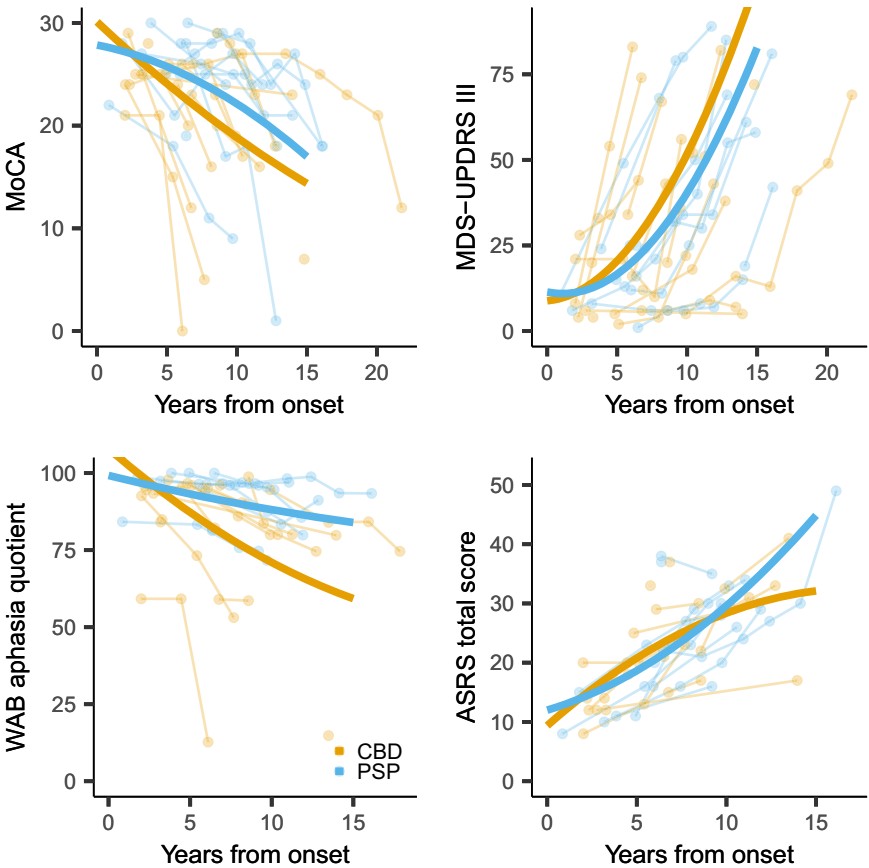

**Fig. 2 Trajectories of cognitive decline, development of parkinsonism and global aphasia, and worsening of apraxia of speech over time from disease onset by pathology.** Cognitive impairment (top left plot) is measured by the Montreal Cognitive Assessment Battery (MoCA); parkinsonism (top right plot) is measured by the Movement Disorders Society-sponsored revision of the Unified Parkinson's Disease Rating Scale part III (MDS-UPDRS III); global aphasia severity (bottom left plot) is measured by the Western Aphasia Battery (WAB) Aphasia Quotient; and apraxia of speech (bottom right plot) by the Apraxia of Speech Rating Scale (ASRS). Individual patient trajectories are shown as well as nonlinear trend lines for both PAOS-CBD and PAOS-PSP patients. Source data are provided as a Source Data file.

**Longitudinal neuroimaging models**. The probabilities that PAOS-CBD and PAOS-PSP differ at baseline, 4 years from baseline, and in rate of change for each imaging modality are shown in Supplementary Figs. 2–4. For FDG no strong evidence (probabilities ≥ 0.99) existed for any differences in regional metabolism at baseline; however, 4 years later, there was strong evidence that PAOS-CBD had lower metabolism in nearly all cortical regions. Rates of decline in metabolism were greater in PAOS-CBD compared to PAOS-PSP in these same cortical regions (examples shown in Fig. 5).

For MRI, there was strong evidence that PAOS-CBD had smaller volumes of striatum at baseline compared to PAOS-PSP, with these regions, plus thalamus, also showing strong evidence for smaller volumes 4 years later (examples shown in Fig. 5). Conversely, there was strong evidence that PAOS-PSP had smaller volumes of cerebellar dentate and midbrain at baseline and 4 years later compared to PAOS-CBD (Fig. 5). There was strong evidence for greater rates of atrophy in striatum, and thalamus in PAOS-CBD and for greater rates of atrophy in cerebellar dentate in PAOS-PSP.

The DTI models focused on FA, since this metric gave the most significant findings in the voxel-based analyses. No strong evidence for any differences in regional FA between groups appeared at baseline; however, 4 years later, there was strong evidence that PAOS-CBD had lower FA in the superior and posterior corona radiata and middle cerebellar peduncle compared to PAOS-PSP (example shown in Fig. 5). There was no strong evidence of differences in rates of decline of FA.

**Neuroimaging correlations with clinical measures**. Worse performance on the Token Test correlated with lower FDG-PET metabolism in Broca's area and left superior temporal gyrus in PAOS-CBD, with similar nonsignificant trends in PAOS-PSP (Supplementary Fig. 5). There was a trend for worse performance on WAB-AQ to correlate with lower metabolism in Broca's area in PAOS-CBD. No regional correlations were identified with ASRS.

**Relationship between regional pathology and AOS subtype**. The CBD-phonetic and PSP-phonetic groups showed the highest neuronal loss in the neocortex but relatively lower neuronal loss in the substantia nigra, subthalamic nucleus, and globus pallidus compared to the prosodic groups (Figs. 6 and 7). On the other hand, the CBD-prosodic and PSP-prosodic groups showed lower neuronal loss in the neocortex but higher neuronal loss in the substantia nigra, subthalamic nucleus and globus pallidus (Figs. 6 and 7). Tau lesion counts across all four groups appeared to be highest in corticostriatal regions compared to the subcortical regions (Fig. 6). There was no discernable pattern of tau lesion counts associated with CBD versus PSP or AOS subtypes. When we assessed the corticostriatal to pallido-nigro-luysial (PNL) ratio for neuronal loss, there was separation by AOS subtype independent of pathology, with the phonetic subtype having higher corticostriatal-to-PNL ratio, while the prosodic subtype had a higher PNL-to-corticostriatal ratio (Fig. 6). There was no association between AOS subtype and tau lesion count ratio.

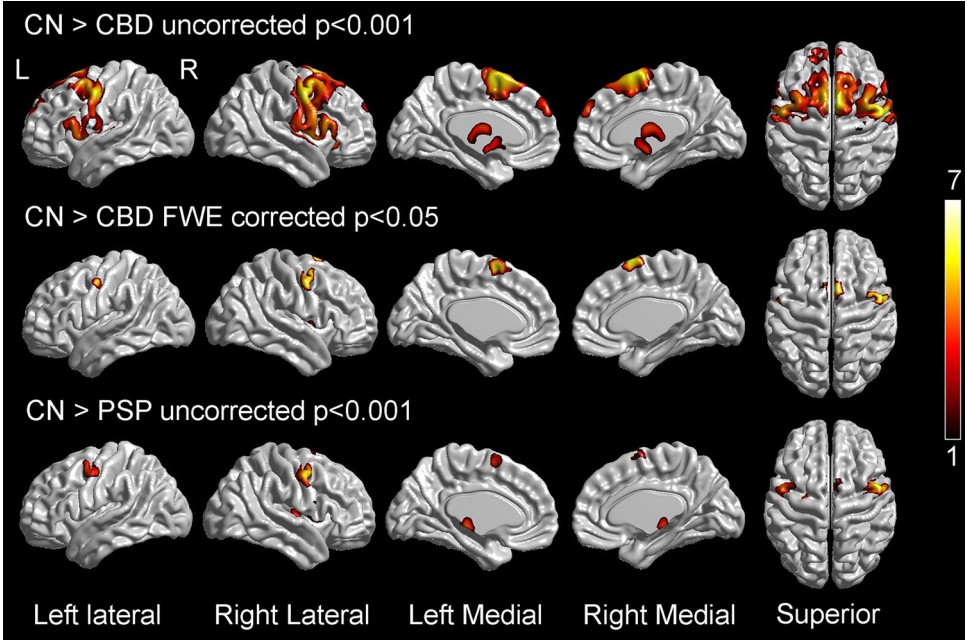

**Fig. 3 Voxel-level FDG-PET findings at baseline.** The top two rows show regions of hypometabolism in progressive apraxia of speech patients with corticobasal degeneration (CBD) compared to controls, either uncorrected for multiple comparisons at $p < 0.001$ (top) or after correction for multiple comparisons using family-wise error (FWE) correction at $p < 0.05$ (middle). The bottom row shows regions of hypometabolism in progressive apraxia of speech patients with progressive supranuclear palsy (PSP) compared to controls shown uncorrected for multiple comparisons at $p < 0.001$. Statistical comparisons were performed using two-sided $t$-tests in SPM12. Color scale reflects $T$ score. Results are shown on three-dimensional brain renderings using BrainNet Viewer.

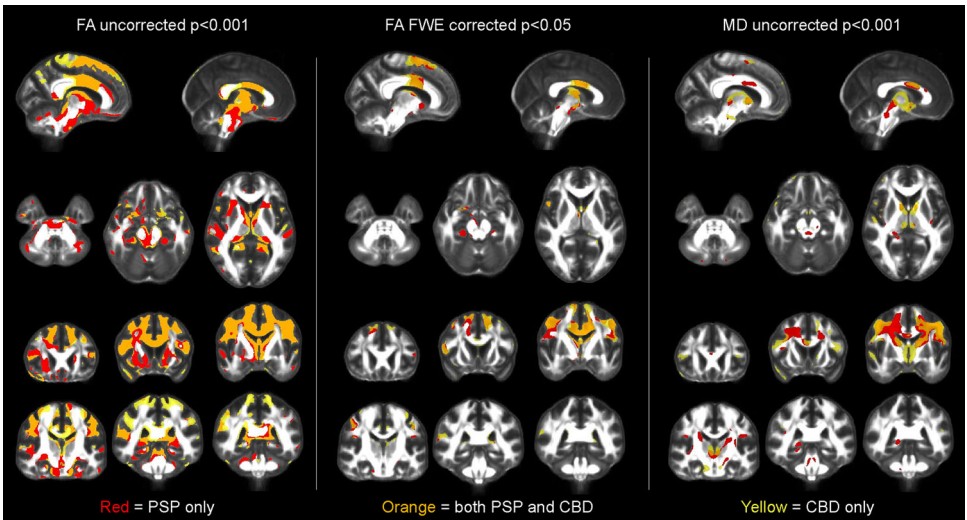

**Fig. 4 Diffusion tensor imaging findings at baseline.** The left two panels show regions with decreased fractional anisotropy (FA) compared to controls, either uncorrected for multiple comparisons at $p < 0.001$ (left) or after correction for multiple comparisons using family-wise error (FWE) correction at $p < 0.05$ (middle). The right panel shows regions with increased mean diffusivity (MD) compared to controls shown uncorrected at $p < 0.001$. Results for progressive apraxia of speech patients with progressive supranuclear palsy (PSP) are shown in red, results for progressive apraxia of speech patients with corticobasal degeneration (CBD) are shown in yellow, and regions of overlap between the two groups are shown in orange. Statistical comparisons were performed using two-sided $t$-tests in SPM12.

**Relationship between regional volume loss and metabolism and AOS subtype**. We observed similar trends in neuroimaging utilizing the last scan before death. The prosodic patients showed more involvement of PNL regions with smaller volumes of globus pallidus, subthalamic nucleus, and substantia nigra compared to phonetic patients. The phonetic patients tended to show lower metabolism in the SMA and motor cortex and smaller volumes of striatum compared to prosodic patients (Supplementary Fig. 6).

**Tau biochemistry**. Examination of the insoluble tau subunits forming the filamentous aggregates in PAOS revealed no specific changes distinct from disease controls (Fig. 8). More specifically, the tau isoform and degradation products resulting from endogenous proteolysis in PAOS appeared identical to the CBS-CBD and PSP with Richardson's syndrome. This was verified by reprobing the CBD and PSP blots with PHF1 antibody to the core of the tau filaments (Supplementary Fig. 7) showing the classic

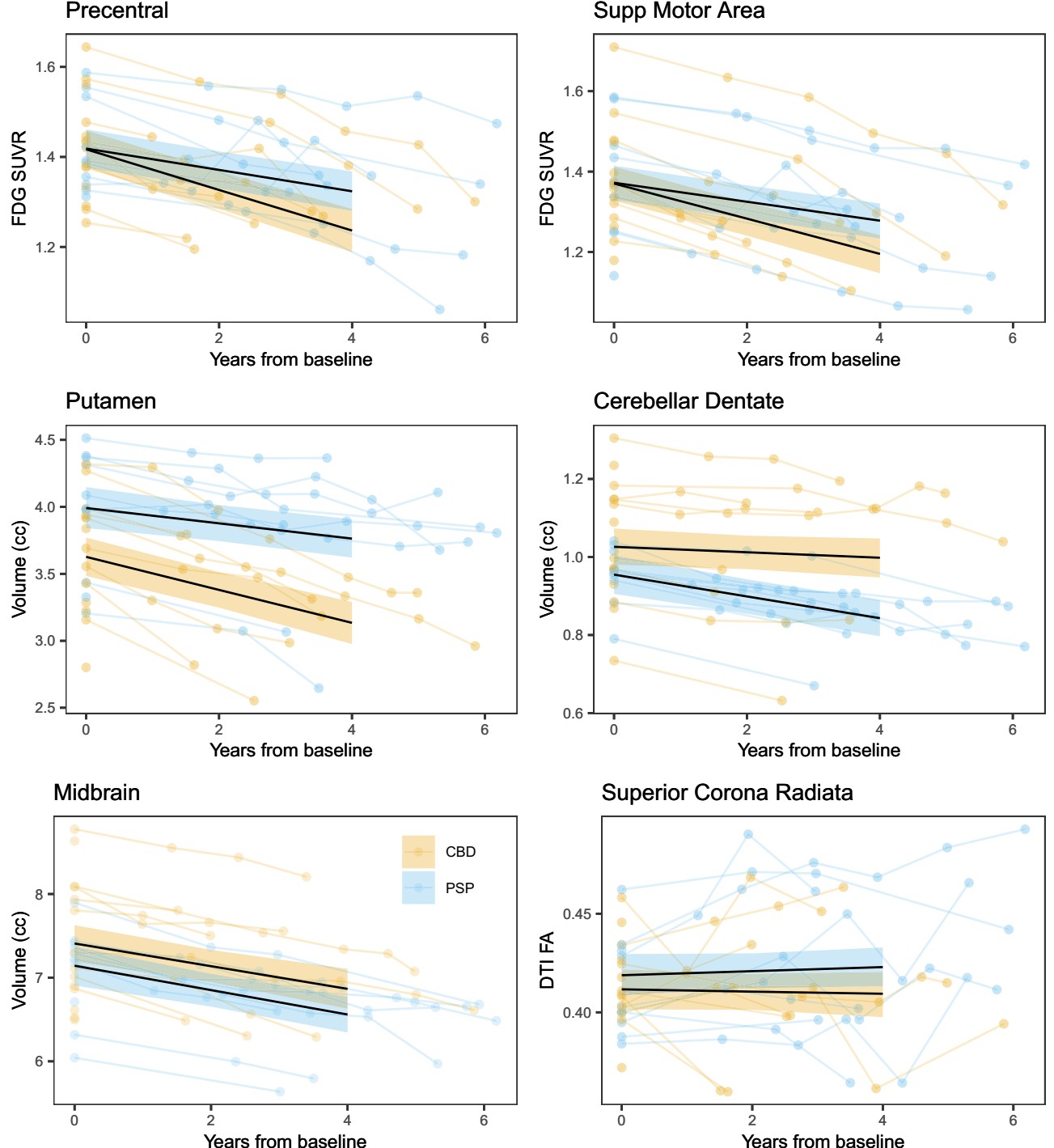

**Fig. 5 Trajectories of decline in regional FDG-PET metabolism, MRI volume, and DTI-fractional anisotropy by pathology.** Trajectories are shown for progressive apraxia of speech patients with corticobasal degeneration (CBD) or progressive supranuclear palsy (PSP) pathology. Median posterior fits of the Bayesian hierarchical models using baseline as the timescale anchor are shown as black lines, with 90% posterior intervals colored by pathological diagnosis for six selected regions, results averaged across hemispheres, across the three neuroimaging modalities. The data used to fit the model are underlaid, again colored by diagnosis, and averaged across hemispheres, with scans from an individual connected by line segments. Fitted values extend to four years, the median time from baseline to death in the CBD group, and where we have a majority of our data across both diagnosis groups. Precentral cortex = motor cortex. Source data are provided as a Source Data file.

~37 kDa CBD doublet or ~33 kDa PSP singlet for filaments having the same morphology.

**Clinical and imaging characteristics of the +AOS patients.** Five patients were categorized as having +AOS (Table 2). All five had

the phonetic subtype of AOS and 4 (80%) were young. The FTLD-TDP type A and PiD cases were all noted to have executive dysfunction or behavioral changes, or both, at presentation. Both FTLD-TDP type A patients had the H1H2 haplotype and mutations in the *GRN* gene (c.1A > C and c.709-2A > G, respectively), and both showed asymmetric hypometabolism in

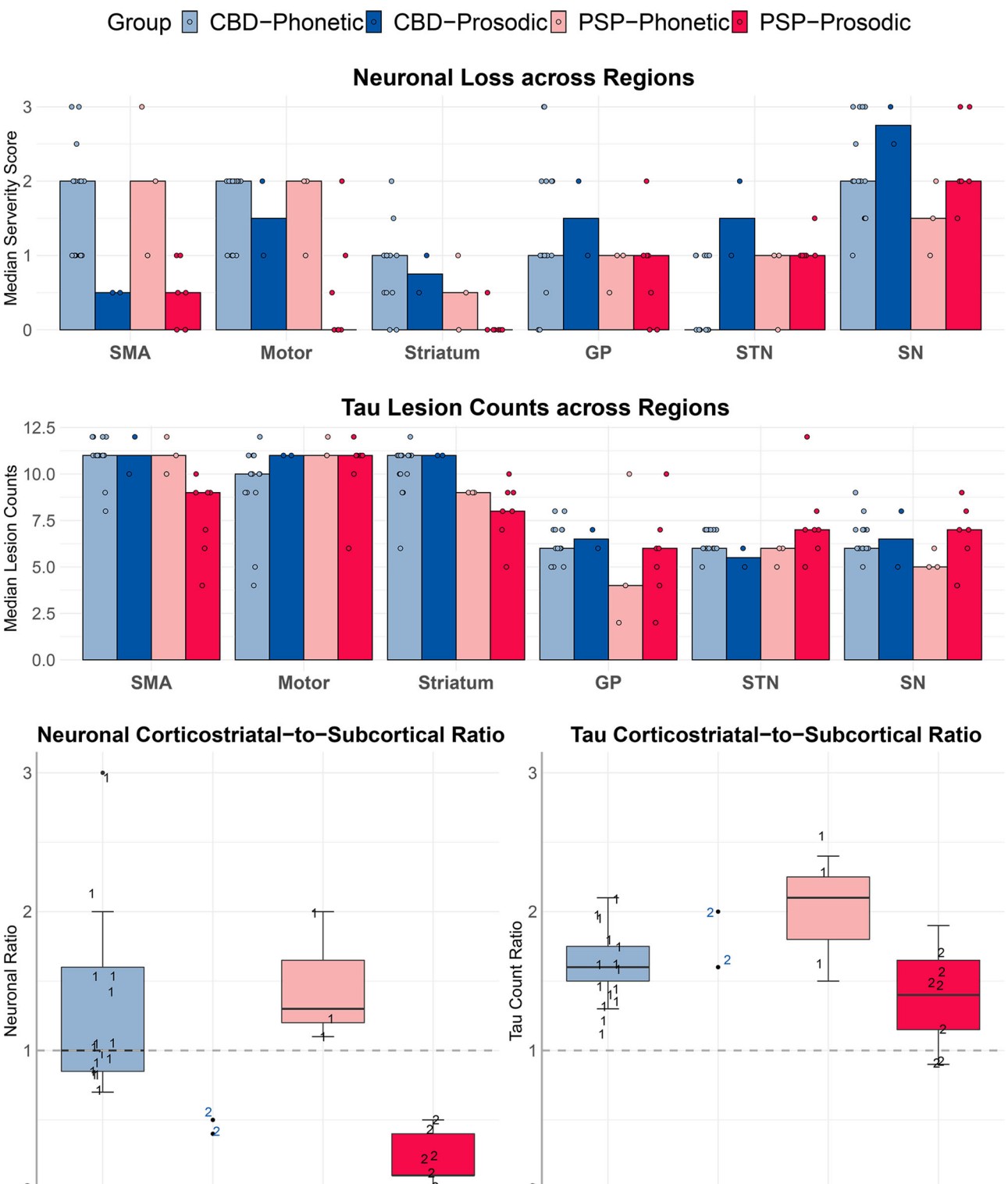

**Fig. 6 Histograms of regional neuronal loss and tau counts by pathology and apraxia of speech subtype.** The top two histogram plots depict median neuronal loss severity or median total tau lesion count for each region, with all individual points shown. The bottom boxplots depict the ratio between corticostriatal regions (supplementary motor area (SMA), motor, striatum) and subcortical structures (globus pallidus (GP), subthalamic nucleus (STN), and substantia nigra (SN)) for both neuronal loss severity and tau lesion count. The phonetic patients are shown with the number 1 and the prosodic patients are shown with a number 2. Boxes represent lower quartile, median and upper quartile, with whiskers extending to the farthest point at most 1.5*inter-quartile range from each quartile. CBD corticobasal degeneration; PSP progressive supranuclear palsy. $N = 15$ CBD-phonetic cases; $N = 2$ CBD-prosodic cases; $N = 3$ PSP-phonetic cases; $N = 7$ PSP-prosodic cases. Source data are provided as a Source Data file.

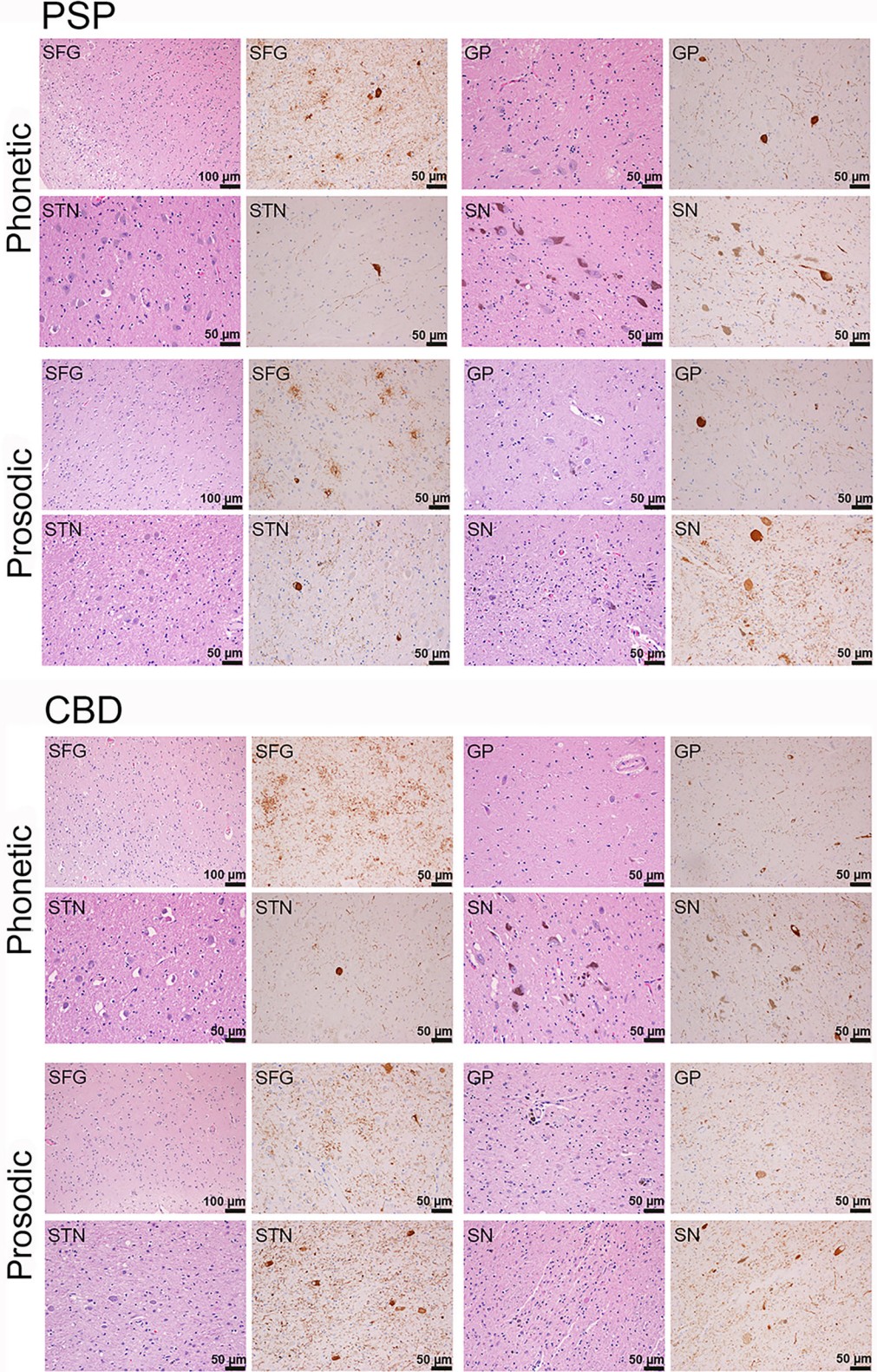

temporoparietal and frontal cortices (Supplementary Fig. 8). Of the two PiD cases, one did not have the expected lobar (knife edge) atrophy (Supplementary Fig. 1). However, both showed frontal hypometabolism (Supplementary Fig. 8). The fifth patient had dysarthria, facial and limb weakness, and mixed pathologies including AGD and co-existing TDP-43 without loss of Betz cells

or medullary motor neurons (CN XII)[33]. The distribution of TDP-43 was widespread involving frontotemporal cortex and brainstem consistent with FTLD-TDP type B while AGD focally involved limbic regions (Supplementary Table 4). This patient showed left anteromedial temporal and posterior frontal hypometabolism (Supplementary Fig. 8).

**Fig. 7 Histological findings of neuronal loss and tau lesion burden in phonetic and prosodic progressive supranuclear palsy (PSP) and corticobasal degeneration (CBD) cases.** PSP-phonetic showed a relatively greater neuronal loss in neocortex (superior frontal gyrus (SFG) compared to subcortical regions (globus pallidus (GP), subthalamic nucleus (STN) and substantia nigra (SN), i.e. pallido-nigro-luysial network). In PSP-prosodic an opposite pattern was observed with relatively greater neuronal loss in pallido-nigro-luysial network (often with pigment and spheroids) compared to neocortex. Differences in the pattern of tau burden were not as striking, although there was subtle evidence for threads to be greater in the neocortex in PSP-phonetic compared to PSP-prosodic. The pattern of neuronal loss in CBD mirrored what was observed in PSP. ×20 magnification. The semi-quantitative regional lesion counts were performed at the time of original histological analysis and independently repeated months to years later for each case at a second time-point by the same investigator (DWD).

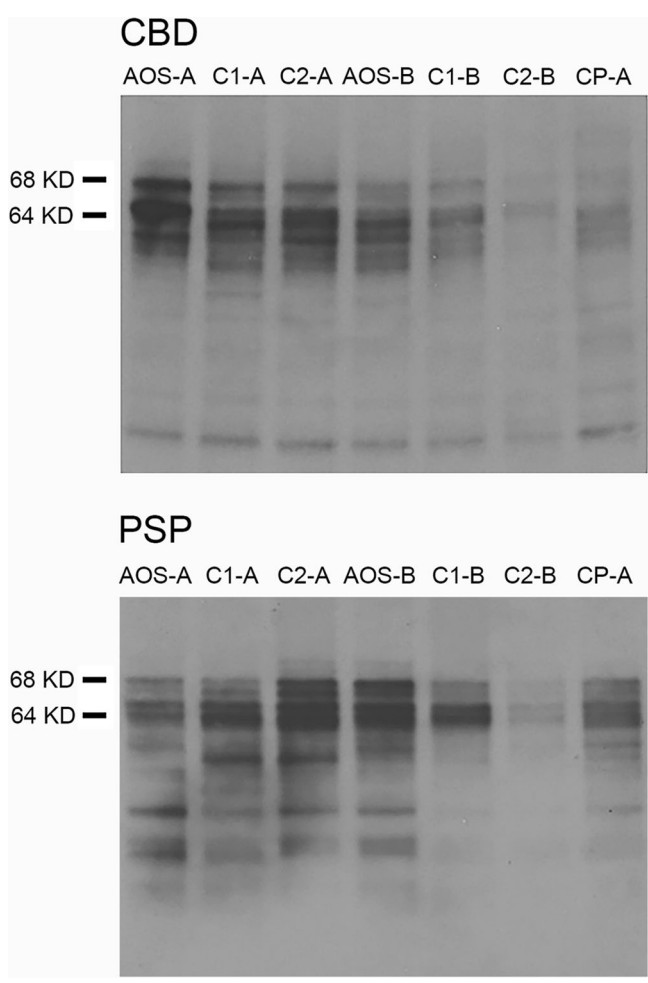

**Fig. 8 Tau biochemistry findings in representative progressive apraxia of speech patients with either corticobasal degeneration (CBD) or progressive supranuclear palsy (PSP) pathology, and controls.** Western blot findings in representative CBD (top panel) and PSP (bottom panel) cases. Apraxia of speech (AOS) denotes the progressive apraxia of speech cases; C1/2 denotes the matched controls; and CP-A denotes pure controls with no co-pathologies. This experiment was replicated independently in eight CBD cases and seven PSP cases with similar results. Two representative cases for each pathology are shown in this figure. The full scan images are provided in Supplementary Fig. 10.

## Discussion

The results of this study offer unique insights into the neuro-biology of PAOS that have important implications for patient prognosis and future development of experimental therapeutic paradigms. We found that PAOS is most often associated with a 4R tauopathy when it exists in isolation or when it co-exists with aphasia. On the contrary, when AOS accompanies behavioral dyscontrol, executive dysfunction, or features of motor neuron disease, the underlying pathology is heterogeneous, less

commonly 4R tau, and may be related to a mutation in the *GRN* gene. We also found that AOS subtype is an important consideration in PAOS; subtype is strongly associated with age at onset, and AOS subtype and age together are highly predictive of pathology. It appears that the different AOS subtypes are linked to different relative degrees of neuronal loss and gliosis in frontal neocortex versus the PNL network. Furthermore, tau haplotypes differed between PAOS-CBD and PAOS-PSP, suggesting that PAOS-PSP, in particular, may have a unique genetic risk profile, albeit with similar tau biochemistry. Finally, neuroanatomical differences were observed between PAOS-CBD and PAOS-PSP on neuroimaging, many of which became more striking with disease progression.

In this prospectively recruited cohort, we found that the most frequent pathology was CBD, occurring in approximately half of the patients. PSP, another 4R tauopathy, was also relatively common, occurring in about one-third. In fact, 100% of our PPAOS and AOS-PAA cohorts had an underlying 4R tauopathy. Indeed, PSP and CBD have been associated with PAOS in case studies and small retrospective series[4,6,19,34]. We also identified other pathologies in our cohort, including PiD, FTLD-TDP type A associated with *GRN* mutations, and FTLD-TDP type B with AGD (a 4R tauopathy), which were all associated with +AOS. The findings of behavioral dyscontrol and executive dysfunction, in addition to AOS, were particularly associated with PiD and FTLD-TDP type A. Indeed, the behavioral variant of FTD is the most common presentation of PiD[35] and FTLD-TDP type A secondary to a *GRN* mutation[36]. These results suggest that if AOS is accompanied by behavioral dyscontrol and executive dysfunction, FTLD-TDP type A and PiD are more likely to be the underlying cause than a 4R tauopathy. Furthermore, given that 50% of these patients had a mutation in *GRN*, such patients should be screened for a mutation in the *GRN* gene. Other likely cases of +AOS reported in the literature similarly lacked 4R tauopathy[8,37].

All patients presenting with PPAOS or AOS-PAA had an underlying 4R tauopathy, with no suggestion that the frequency of these two clinical syndromes differed by the specific 4R tauopathy. On the other hand, AOS subtype and age differed between PAOS-CBD and PAOS-PSP. PAOS-CBD was associated with phonetic AOS and a younger age at onset, while PAOS-PSP was associated with prosodic AOS and older age at onset. This finding provides support for the predictive value of AOS subtype, the importance of which was stressed by Mailend and Maas[38]. Despite the strong relationship between AOS subtype and pathology, there were exceptional cases of PSP presenting with phonetic AOS (3/10) and CBD presenting with prosodic AOS (2/17). On further investigation, there appears to be a histologic explanation for these departures, which may, in fact, shed light on the pathological underpinnings of AOS subtype. We found that the relative amount of neuronal loss affecting the corticostriatal network (SMA/motor cortex and striatum), compared to the amount of neuronal loss affecting the PNL network (globus pallidus, substantia nigra, and subthalamic nucleus), differed by AOS subtype. The phonetic CBD and PSP patients showed

**Table 2 Baseline demographic and clinical features for the five +AOS cases with FTLD-TDP, PiD, and AGD pathology.**

| | FTLD-TDP A | FTLD-TDP A | FTLD-TDP B + AGD | PiD | PiD |
|---|---|---|---|---|---|
| **Demographics and genetic data** | | | | | |
| Onset to death, years | 5 | 5 | 2 | 8 | 6 |
| Onset to baseline, years | 2 | 3 | 2 | 4 | 2 |
| FTD mutations | *GRN* | *GRN* | None | None | None |
| *APOE* genotype | 3/4 | 3/3 | 3/3 | 3/3 | 2/3 |
| H1 haplotype | H1/H1 | H1/H2 | H1/H1 | H1/H1 | H1/H2 |
| Family history in 1st or 2nd degree relative | None | None | None | Present | Present |
| *TMEM106B* CC/CG/ GG | CG | CG | CC | CC | GG |
| **Neurological data** | | | | | |
| Presenting symptoms | (Executive dysfunction, parkinsonism) + AOS | (Mild behavioral changes, aphasia) + AOS | (Dysarthria, dysphagia, face/tongue weakness anomia) + AOS | (Mild behavioral changes, aphasia) + AOS | (Obsessive compulsive behaviors) + AOS |
| Final diagnosis before death | CBS | Probable bvFTD | Unclassified | Probable bvFTD | Possible bvFTD |
| MMSE, /30 | 30 | 13 | 24 | 29 | 27 |
| MoCA, /30 | 28 | 17 | 24 | 29 | 22 |
| FAB, /18 | 18 | 8 | 15 | 18 | 15 |
| FBI, /72, 0 = best | 4 | 21 | 8 | 28 | 7 |
| NPI-Q, /36 | 0 | 7 | 7 | 2 | 5 |
| MDS-UPDRS III, /132, 0 = best | 9 | 5 | 24 | 7 | 11 |
| WAB Praxis, /60 | 60 | 49 | 57 | 57 | 58 |
| PSIS, /5, 0 = best | 0 | 1 | 0 | 0 | 0 |
| **Speech and language data** | | | | | |
| AOS subtype | Phonetic | Phonetic | Phonetic | Phonetic | Phonetic |
| Aphasia present | No | Yes | Yes | Yes | No |
| Aphasia severity /4 | 0 | 2 | 1 | 1 | 0 |
| WAB Aphasia Quotient, /100 | 97 | 78 | 95 | 96 | 96 |
| ASRS Total Score V3 | 10 | 11 | 10 | 14 | 6 |
| ASRS Phonetic subscore | 6 | 5 | 4 | 7 | 4 |
| ASRS Prosodic subscore | 0 | 2 | 2 | 2 | 0 |
| MSD Severity Scale, /10 | 6 | 5 | 7 | 6 | 9 |
| BNT, /15 | 14 | 13 | 10 | 13 | 14 |
| WAB fluency, /10 | 9 | 5 | 9 | 10 | 9 |
| Letter fluency (FAS) | 46 | 12 | 14 | 21 | 16 |
| WAB animal fluency | 20 | 13 | 11 | 20 | 13 |
| Token Test, /22 | 19 | 4 | 19 | 21 | 17 |
| NAT, /10 | 9 | 6 | 10 | 8 | 9 |
| PPT word-word, /52 | 52 | 41 | 50 | 49 | 51 |
| Dysarthria Present, N % | No | No | Yes | No | No |
| Dysarthria Severity, /4 | 0 | 0 | 1 | 0 | 0 |
| NVOA, /32, 32 = normal | 31 | 17 | 31 | 14 | 21 |
| **Neuropsychological data** | | | | | |
| TMT A MOANS | 11 | 5 | 13 | 11 | 7 |
| TMT B MOANS | 7 | 3 | 9 | 12 | 3 |
| Sorting Test SS | 12 | 0 | 11 | 13 | 3 |
| WMS-III VR I SS | 7 | 5 | 9 | 8 | 2 |
| WMS-III VR II SS | 9 | 9 | 8 | 13 | 5 |
| VOSP letters, /20 | 20 | 19 | 20 | 20 | 19 |
| VOSP cubes, /10 | 10 | 5 | 10 | 8 | 6 |

MOANS and SS have a mean of 10 with a standard deviation of 3 in normal healthy individuals.

*APOE* apolipoprotein, *GRN* progranulin, *MAPT* microtubule associated protein tau, *MMSE* Mini Mental State Examination, *MoCA* Montreal Cognitive Assessment, *FAB* Frontal Assessment Battery, *FBI* Frontal Behavioral Inventory, *NPI-Q* short questionnaire version of the Neuropsychiatric Inventory, *MDS-UPDRS III* Movement Disorder Society sponsored revision of the Unified Parkinson's Disease Rating Scale part III, *WAB* Western Aphasia Battery, *PSIS* Progressive supranuclear palsy Saccadic Impairment Scale, *AOS* apraxia of speech, *ASRS* Apraxia of Speech Rating Scale, *MSD* Motor Speech Disorder, *BNT* Boston Naming Test, *NAT* Northwestern Anagram Test, *PPT* Pyramids and Palm Trees test, *NVOA* nonverbal oral apraxia, *SS* scaled score, *TMT* Trail Making Test, *WMS-III VR* Wechsler Memory Scale-III Visual Reproduction, *MOANS* Mayo Older American Normative Studies, *VOSP* Visual Object and Space Perception Battery.

relatively greater involvement of the corticostriatal network compared to the PNL network, while the prosodic PSP and CBD patients showed the opposite pattern. This finding supports regions in the corticostriatal network being involved with programing and planning of motor speech and, hence, linked to dysfunction of motor speech output such as distorted sound substitutions, additions, and prolongations. The phonetic features of AOS also occur in the context of stroke, and in such cases, lesion analysis has implicated the premotor and motor cortices[39,40]; resting-state fMRI studies have also implicated the left premotor cortex[41]. In the context of PPAOS, we have also found more widespread involvement of premotor and motor cortices on neuroimaging in phonetic AOS[12]. The five +AOS cases all had a phonetic AOS associated with pathologies that targeted the cortex over the PNL network. The PNL network has not previously been implicated in AOS. These regions are commonly associated with motor parkinsonism, progressive akinesia in which slowing is a pathognomonic feature, as well as freezing of gait[42]. On the other hand, there is literature linking the PNL network to speech. In a study of PSP patients with prominent involvement of the PNL network, motor speech problems occurred as an early feature in 100% of the patients[28]. Although it is not proven that PNL network-related speech problems are equivalent to prosodic AOS, they are commonly described as an akinesia of speech (lack of, or slowing); this could represent a variant of prosodic AOS, given that slow rate is one of its defining features. Severe degeneration of the substantia nigra (without pathological description of globus pallidus and subthalamic nuclei) at autopsy has also been reported in a PAOS patient who subsequently developed parkinsonism[43]. In support of the pathological associations, we also found neuroimaging evidence for differences in the corticostriatal and PNL networks with phonetic and prosodic AOS. We acknowledge that our neuroimaging findings were not as robust as the pathological findings, possibly due to lack of precision in the large cortical ROIs and difficulty measuring small subcortical structures. Nevertheless, this finding suggests potential value of future interrogation of these networks using connectivity-based analyses and imaging techniques that allow more precise measurement of these regions.

Beyond age and AOS subtype, there were few differences in clinical features between PAOS-CBD and PAOS-PSP at baseline that would help to differentiate between them. Longitudinally, PAOS-CBD showed an overall faster rate of decline in cognitive and language function yet showed a similar rate of worsening in AOS severity compared to PAOS-PSP. We found that aphasia severity showed the expected correlations with metabolism in language areas and these correlations were stronger in PAOS-CBD, which fits with the fact that PAOS-CBD patients more often developed aphasia.

On neuroimaging, baseline differences between PAOS-CBD and PAOS-PSP were apparent on MRI. Specifically, PAOS-CBD showed smaller volumes of the striatum, and PAOS-PSP showed smaller volume of the cerebellar dentate and midbrain. These differences persisted 4 years after onset, with additional involvement of thalamus in PAOS-CBD. Involvement of the cerebellar dentate and midbrain are characteristic features of Richardson's syndrome[44], and we have previously shown that midbrain atrophy can develop in PAOS[13], especially in those patients who develop Richardson's syndrome[16]. It is difficult to know whether the presence of cerebellar dentate and midbrain atrophy in our PAOS cohort represents a marker of PSP pathology[45] or simply reflects the fact that the PAOS-PSP patients commonly developed Richardson's syndrome[16]. The finding of greater striatal and thalamic atrophy in PAOS-CBD compared to PAOS-PSP is consistent with one previous study[34] and with the association of CBS and CBD with atrophy of the basal ganglia[46]. No differences

were observed in the neocortex between PAOS-CBD and PAOS-PSP at baseline with FDG-PET; patterns of hypometabolism were consistent with our previous work on PAOS[11]. Differences between pathologies became apparent, however, as the disease progressed, with PAOS-CBD showing faster rates of decline across most neocortical regions compared to PAOS-PSP. The presence of widespread cortical neurodegeneration in PAOS could, therefore, be an indicator of underlying CBD pathology.

A large degree of overlap was observed across PAOS-CBD and PAOS-PSP in patterns of white-matter degeneration, predominantly involving frontal and motor cortex white matter, body of the corpus callosum, cingulum, and superior longitudinal fasciculus. This network of regions has previously been shown to be abnormal in PAOS[13,47]. We observed no significant differences between pathologies at baseline, and little difference was observed across pathologies in the rate of decline in white matter integrity; however, as the disease progressed, PAOS-CBD had greater degeneration of the corona radiata compared to PAOS-PSP. This is consistent with the fact that the corona radiata receives and sends projections to the cortex, which is particularly affected in PAOS-CBD. The corona radiata is frequently affected in CBS [48].

Homozygosity of the *MAPT* H1 allele (i.e., H1H1) increases the risk of PSP and CBD[20,21], while a genome-wide study suggested that the presence of the H2 allele is protective[49]. Our study found the expected high frequency of the H1 allele in our PAOS-CBD patients. Surprisingly, however, we found that PAOS-PSP had a lower frequency of the H1 allele compared to PAOS-CBD and also compared to our large pathologically-confirmed cohort with typical Richardson's syndrome, suggesting the *MAPT* H1 allele may not be a risk for PAOS-PSP. Another possibility is that the higher frequency of the H2 haplotype within PAOS-PSP cases could be conferring a predisposition of the PAOS-PSP phenotype on carriers as opposed to Richardson's syndrome.

We did not find any differences in the insoluble tau subunits forming the filamentous aggregates in our PAOS-CBD and PAOS-PSP cases. Indeed, the tau isoform composition was consistent with CBD and PSP, and most importantly, the degradation products resulting from endogenous proteolysis for these cases were similar, indicating the underlying pathobiology of tau, in contrast to neuroanatomy of degenerative changes, is not different in patients with AOS[22]. Hence, there is no biochemical evidence to suggest that PAOS-CBD and PAOS-PSP represent distinct tauopathies.

There was no enrichment for the *TMEM106B* CC genotype in our PAOS cohort as was recently described in cases with an atypical tauopathy[50]. The frequency of the C and G allele in our PAOS cohort was similar to what we previously reported in neurologically healthy controls, although there was a slightly lower frequency of the GG genotype, which is thought to be protective against FTLD-TDP type A and *GRN* mutations when compared to controls (PAOS = 31.2% CC, 59.4% CG, 9.4% GG; healthy controls = 32.7% CC, 48.1% CG, 19.1% GG controls) [51].

Strengths of our study include the fact that it is a longitudinal (10-year) NIH-funded prospective study, all patients underwent detailed and standardized clinical batteries at each visit, and multiple different analytic techniques complimented and supported one another. Limitations include the relatively small size of our +AOS group and the lack of racial diversity in our patient cohort.

The findings from this study have important implications for prognosis, management, and counseling patients and families about the disease course. For example, phonetic patients will more likely lose the use of their limbs, while prosodic patients will more likely develop poor balance, fall, lose the ability to walk, and be wheelchair bound. Targeted physical therapy will therefore differ somewhat in the phonetic versus prosodic patients. There

are also implications for speech-language therapy and management strategies including aiding with long-term planning regarding employment, travel, and the form of alternative means of communication that may eventually be required. Specifically, PAOS patients without aphasia have a slower course, longer survival, and better prospects for maintaining sufficient language skills for day-to-day communication and handling work demands than patients who are also aphasic; phonetic patients have faster rates of decline in language ability and may be at greater risk for the emergence of behavioral and other cognitive problems. Alternative communication strategies may be more effective and easier to implement in the absence of aphasia. Last but not least, our findings also have implications on the design of clinical trials of therapies targeting 4R tauopathies, influencing mechanisms for targeting individuals for enrollment, selecting appropriate imaging biomarkers, and developing therapeutic paradigms to target specific brain networks. This could pertain to differences in regional targeting or neurochemical targeting, since neurotransmitters in cortical network differ from those in the PNLA network.

## Methods

**Patient recruitment**. The Neurodegenerative Research Group (NRG), Mayo Clinic, recruited patients who presented to the Department of Neurology with an AOS suspected to be secondary to a degenerative process between 7/1/2010 and 12/31/2020 (Supplementary Fig. 9). That is, the speech problem was insidious in onset and progressive in nature. Only patients over age 18 with an informant to provide an independent evaluation of functioning, and who spoke English as their primary language (including bilingual patients), were enrolled; none were 18 or younger. All patients underwent detailed neurological evaluation, speech and language examination, neuropsychological testing, blood sampling for DNA extraction, and multimodal neuroimaging that included a 3.0 Tesla volumetric head MRI scan, diffusion tensor imaging (DTI), and [18F] fluorodeoxyglucose positron emission tomography (FDG-PET) scan, over a span of 48–72 h. Family history was abstracted, and positive family history was defined as history of any neurodegenerative disease (primary progressive aphasia, frontotemporal dementia, progressive apraxia of speech, PSP, CBS, Alzheimer's disease <age 65, amyotrophic lateral sclerosis (ALS), Parkinson's disease, dementia with Lewy bodies) in a first- or second-degree relative. All patients were followed longitudinally with yearly evaluations with identical clinical and multi modal neuroimaging. For this study, we included all PAOS patients who have subsequently died ($n = 32$). Some of these patients have been included in previous clinical studies[9–13,15,18], including 10 that were in our original manuscript describing AOS subtypes (these patients are highlighted in Supplementary Fig. 9)[15]; pathology has only been reported in a single patient from this cohort[15]. There was no evidence for a bias in the length of follow-up time or time from onset to last visit between the entire cohort and the 32 patients that had an autopsy (Supplementary Fig. 9).

This study complied with all relevant ethical regulations for work with human participants and was approved by the Mayo Clinic Institutional Review Board (Clinical trial registration numbers: NCT01818661; NCT03313011; NCT01623284). Prior to death, all patients or their proxies had provided written informed consent for research, including brain autopsy examination, which was reaffirmed upon death. All clinical data were collected in a REDCap database, version 10.0.28.

**Speech and language examination and diagnosis**. The speech and language protocol included the following battery of tests: the WAB revised[52] to assess global language ability; the Token Test, Part V[53] to assess spoken language comprehension; the Northwestern Anagram Test[54] to assess syntactic ability; the 15-item BNT[55] to assess confrontation naming; the Peabody Picture Vocabulary Test[56] to assess receptive lexicon; and the Motor Speech Disorder severity scale (adapted from Yorkson et al.[57]) to assess motor speech severity. Letter fluency (FAS) and semantic (animal) fluency scores were also obtained[58]. The ASRS version 3.0[59] was utilized to assess AOS characteristics and severity, which included sub-scores that measured phonetic and prosodic characteristics. Judgments about motor speech abilities were based on all spoken language tasks of the WAB plus additional speech tasks that included vowel prolongation, speech-like alternating motion rates, speech-like sequential motion rates, word and sentence repetition tasks, and a conversational speech sample. The same speech tasks were also judged for the presence or absence of dysarthria, which was rated on a 0–4 severity scale. Judgments about the presence of aphasia were based on performance on all measures of spoken and written language. Aphasia severity was graded on a 0–4 scale based on clinical judgment. To meet the criteria for having agrammatism, function word omissions or syntactic errors had to be present on at least two tasks including but not limited to spontaneous speech in general conversation or during verbal or written examination tasks. Nonverbal Oral Apraxia was assessed using an eight-item scale[17].

The patient diagnosis was made after review of video and audio recordings of speech and language test scores by consensus between two or three speech-language pathologists (J.R.D., R.L.U., H.M.C., and E.A.S.) blinded to all neurological, neuropsychological, and imaging findings. Patients with PAOS in its relatively pure manifestation, i.e., in the absence of aphasia or any neurological/neuropsychological signs or symptoms determined by history or formal testing (see below), were designated as PPAOS. Patients with PAOS and evidence of progressive agrammatic aphasia were designated as AOS-PAA. Patients with PAOS co-existing with one or two other neurological signs and symptoms, such as behavioral changes, executive dysfunction, parkinsonism, or face/limb weakness, yet not meeting criteria for another neurodegenerative syndrome/disease (such as PSP, CBS, ALS, etc.), were designated as +AOS. Judgment of AOS subtype was based on characteristics during spontaneous speech, and structured speech tasks and judgments were made blinded to all neurological, neuropsychological and imaging findings[11,12]. A designation of phonetic AOS was made if the predominant characteristics of the AOS were distorted sound substitutions, deletions, or additions. A designation of prosodic AOS was made if the predominant characteristics of the AOS were lengthened inter-segment durations between syllables within words, between words, or both. A designation of mixed AOS was made if features characteristic of phonetic or prosodic were too mild, too severe, or equal in severity. Those who were too mild at baseline but were declared either a phonetic or prosodic AOS subtype ($n = 2$) one year later were categorized with this latter designation for analysis.

**Neurological evaluation**. All patients underwent detailed neurological examination by a behavioral neurologist and/or movement disorders specialist (K.A.J., H.B., F.A.) at each visit. The examination included the Mini-Mental State Examination and MoCA to assess general cognitive function; the Frontal Assessment Battery to assess executive function; the FBI to assess the degree of behavioral dyscontrol; the brief questionnaire form of the Neuropsychiatric Inventory to assess neuropsychiatric features; the Limb Apraxia subscale of the WAB to assess ideomotor apraxia; the MDS-UPDRS Part III to assess motor parkinsonism; the PSP Saccadic Impairment Scale to assess eye movement abnormalities; and documentation of the presence or absence of face or limb weakness, spasticity, or other corticospinal tract signs, myoclonus, and dystonia.

**Neuropsychological evaluation**. The neuropsychological battery was performed by a trained neuropsychometrist and was overseen by a board-certified neuropsychologist (MMM). It included the DKEFS Sorting Test and Trail Making Test (TMT) B to assess executive function; TMT A to assess cognitive speed; the Wechsler Memory Scale-III Visual Reproduction test (I and II) to assess visual memory; and the Visual Object and Space Perception Battery cube analysis and incomplete letters tests to assess visuospatial and visuo-perceptual function[60].

**Neuroimaging**. Our analysis utilized three different imaging modalities: (1) FDG-PET to assess dysfunction in cortical regions; (2) volumetric head MRI to assess volume of subcortical structures; and (3) DTI to assess integrity of the white matter. All patients underwent FDG-PET using a PET/CT scanner (GE Healthcare, Milwaukee, Wisconsin) operating in 3D mode. Patients were injected with average 459 MBq (range, 367-576 MBq) of [18F]FDG, and after a 30-min uptake period, an 8-min FDG scan was performed consisting of four 2-min dynamic frames following a low-dose CT transmission scan. All patients also underwent a standardized MRI imaging protocol at 3.0 Tesla on a GE scanner, which included a 3D MPRAGE sequence and a single-shot echo-planar DTI pulse sequence with 41 diffusion encoding steps and five nondiffusion weighted T2 images. All MPRAGE images underwent pre-processing correction for intensity nonuniformity using SPM12 (revision 8369) (https://www.fil.ion.ucl.ac.uk/spm/).

Group-level analyses focused on assessing neuroimaging in patients with either CBD or PSP pathology, given that these were the most common pathologies underlying PAOS. For FDG-PET, we performed voxel-level and region-of-interest (ROI) level analyses. All FDG-PET scans were registered to the concurrent MPRAGE using 6 degrees-of-freedom registration in SPM12 (revision 8369) with default settings. All voxels in the FDG-PET images were divided by median uptake in the pons to create standardized uptake value ratio (SUVR) images, and these SUVR images were normalized to the Mayo Clinic Adult Lifespan Template (MCALT) (https://www.nitrc.org/projects/mcalt/) version 1.4 using the normalization parameters from the MPRAGE normalization to MCALT[47], and smoothed at 6 mm full-width-at-half-maximum. Voxel-wise t-tests in SPM12 were used for statistical comparisons of baseline FDG-PET for patients and healthy controls with age and gender as covariates. The control individuals ($N = 25$, median [inter-quartile range] age = 69 [62–75] years, 44% females; MMSE range 26–30) were recruited into NRG and age- and gender-matched to the 32 PAOS patients. Results were assessed in SPM12 after family-wise error (FWE) correction for multiple comparisons at $p < 0.05$ or uncorrected at a threshold of $p < 0.001$. The MCALT atlas was also used to output ROI-level SUVR values for cortical regions across the frontal, temporal (lateral only), parietal and occipital lobes, using Advanced Normalization Tools (ANTs) version 1.9.x[61] with MCALT default

settings. ROI-level SUVRs were calculated separately for all serial FDG-PET for each patient. Individual patterns of hypometabolism were assessed using the clinical tool of 3-dimensional stereotactic surface projections (SSP) with the software package CortexID (Suite 2.1; GE Healthcare, Waukesha, Wisconsin). The activity in each patient's PET data set was normalized to the pons and compared with an age-segmented normative database, yielding a 3-dimensional SSP $z$ score image. The image produced by this analysis produces a metabolic map using the $z$ scores as calculated for each surface pixel.

For MRI, SPM12 with the MCALT atlas was used to output volumes of the following, mainly subcortical, structures: amygdala, parahippocampal gyrus, entorhinal cortex, fusiform, hippocampus, caudate, putamen, globus pallidus, thalamus, and cerebellum. In-house developed atlases were used to output volume of the midbrain and dentate nucleus of the cerebellum. These atlases were created by manually drawing these structures onto MCALT. The Deep Brain Stimulation Intrinsic Template Atlas (DISTAL) atlas was used to output volumes of the subthalamic nucleus and substantia nigra. ROI-level volumes were calculated separately for all serial MRI for each patient.

For DTI, we performed voxel-level and ROI-level analyses. The DTI scans were denoised, corrected for head motion, eddy current distortion, Gibbs-ringing, Rician bias, and skull stripped. Diffusion tensors were estimated using nonlinear least squares fitting and used to calculate FA and MD. Each DTI image was nonlinearly co-registered[61] and normalized to a 1-mm isotropic Montreal Neurological Institute 152 standard space via the FMRIB58_FA template using ANTs version 1.9.x with default settings[61]. Regions of cerebrospinal fluid and gray matter were removed from consideration by masking out regions where mean FA across co-registered patient images were below 0.2. A whole-brain voxel-level analysis of baseline FA and MD was performed in SPM12 comparing patients to controls. Results were assessed corrected for multiple comparisons using the FWE correction at $p < 0.05$ and uncorrected at $p < 0.001$, with age and gender included as covariates. ROI-level data was generated using the JHU Eve WM atlas separately for all serial FA images for each patient.

**Neuropathologic evaluations.** All 32 cases underwent neuropathological evaluations by an experienced neuropathologist, in accordance with current diagnostic protocols[62]. Twenty-eight of the 32 cases were autopsied at Mayo Clinic, Jacksonville, FL (DWD), while the remaining four cases were autopsied at Mayo Clinic, Rochester, MN (RRR, $n = 2$) and Northwestern University (EHB, $n = 2$). Histology slides and/or tissue blocks of the four cases autopsied at Mayo Clinic, MN, and Northwestern were sent to Mayo Clinic, FL, for analysis in order to standardize all pathological data.

Immunohistochemistry was performed using a battery of antibodies including antibodies to α-synuclein (rabbit polyclonal [NACP, Mayo Clinic antibody], 1:3000 with 95% formic acid pretreatment and DAKO EnVision reagents [Carpinteria, CA]); phosphorylated TDP-43 antibody (pS409/410, 1:5,000 mouse monoclonal, Cosmo Bio Co., LTD); amyloid-beta (Aβ) (6F/3D, 1:250, human Aβ8–17, DAKO, Carpinteria, CA); phospho-tau (CP13; 1:1000; IgG1 to phospho-serine 202, gift from the late Dr. Peter Davies, Feinstein Institute, Long Island, NY); 4R-tau (RD4, 1:5000, Millipore, Temecula, CA) and 3R-tau (RD3, 1:5000, Millipore, Temecula, CA). Pathological diagnoses were rendered according to published pathological criteria. PiD was diagnosed based on the presence of argyrophilic and tau-immunoreactive Pick bodies[63], which are negative on Gallyas silver stain. PSP was diagnosed based upon presence of characteristic neuronal (pretangles and tangles) and glial lesions (tufted astrocytes and oligodendroglial coiled bodies (CB)) in vulnerable cortical and subcortical regions[64]. PSP was subclassified as typical or atypical based upon departure from expected distribution of tau pathology and neuronal loss[24]. CBD was diagnosed by the presence of cortical and subcortical neuronal and glial lesions (astrocytic plaques) and thread-like processes in gray and white matter[23]. FTLD-TDP type A had TDP-43 immunoreactive neuronal cytoplasmic inclusions and dystrophic neurites, as well as neuronal intranuclear inclusions, in vulnerable cortical and subcortical areas while type B had predominantly neuronal cytoplasmic inclusions[65]. Senile plaques and NFTs were evaluated with thioflavin S fluorescent microscopy, as well as Bielschowsky and Gallyas silver stains. Each case was assigned a Braak NFT stage[66], and Thal phases assessed the presence and distribution of Aβ plaques[27]. If NFTs were detected in the absence of Aβ plaques, a diagnosis of primary age-related tauopathy was rendered. Other concomitant pathologies, such as Lewy body disease, age-related tau astrogliopathy, AGD, hippocampal sclerosis, and vascular disease (cerebral amyloid angiopathy, microinfarcts and large, lacunar and hemorrhagic infarcts) were also recorded[29].

**Semi-quantitative neuropathologic methods.** For all PSP and CBD cases, neuronal loss was assessed on hematoxylin and eosin (H&E) and tau lesion count was assessed using phospho-tau (CP13) immunohistochemistry by DWD. Neuronal loss and semi-quantitative tau lesion counts were assessed in the following regions: *cortical* ($n = 4$, SMA, motor cortex, superior temporal and inferior parietal) and *subcortical* ($n = 4$, striatum, globus pallidus, subthalamic nucleus and substantia nigra). These regions were specifically assessed given their likely involvement in motor speech and or aphasia. Furthermore, in many of these regions, pathology is associated with parkinsonian features. For each region, neuronal loss was scored on a 4-point scale: 0 = absent; 1 = mild, 2 = moderate and 3 = severe. Tau lesion scores were assessed using a semi-quantitative 4-point scale (0 = absent, 1 = mild,

2 = moderate, 3 = severe) for neurofibrillary tangles (NFT + pretangles), oligodendroglial CB, astrocytic plaques (AP for CBD) or tufted astrocytes (TA for PSP), and tau threads. For each of the six regions, a total tau lesion score was calculated based on the summed score of each of the four individual lesion types (NFTs + CB + (TA or AP) + TD). All pathological data were collected in Microsoft Access (15.0).

For case 3, we performed semi-quantitation of both TDP-43 and AGD lesion count across neocortical, limbic, subcortical and brainstem regions as follows: none = 0; scant number of inclusions = 1+, moderate number of inclusions = 2+, frequent number of inclusions = 3+. TDP-43 lesion count based on neuronal cytoplasmic inclusions.

**Genetic screening.** Genomic DNA was extracted from the fresh-frozen brain using the QIAamp DNA mini kit (Qiagen, Leusden, The Netherlands). Amplification was carried out using a modified version of the single-day apolipoprotein ε method[67]. A *MAPT* variant (rs8070723) tagging the H1/H2 haplotype was genotyped using TaqMan SNP genotyping assay ID C_29297996_10 on an ABI QuantStudio 7 Flex Real-Time polymerase chain reaction (PCR) System (Thermo Fisher Scientific, Waltham, MA). Genotypes were called using QuantStudio Real-Time PCR Software (Thermo Fisher Scientific, Waltham, MA). The frequencies of H1/H1, H1/H2, and H2/H2 in the PAOS-CBD and PAOS-PSP cases were compared to the frequencies observed in 802 autopsy-confirmed Richardson's syndrome[32], PSP cases, and 230 autopsy-confirmed CBS[30] CBD cases (CBS-CBD) from the CurePSP Brain Bank located at Mayo Clinic, Jacksonville, FL.

The coding *TMEM106B* variant p.T185S (rs3173615) tagging the *TMEM106B* risk and protective haplotypes was genotyped in all cases using TaqMan SNP genotyping assay ID C_27465458_10 on an ABI QuantStudio 7 Flex Real-Time PCR System according to manufacturer instructions (Thermo Fisher Scientific, Waltham, MA). Genotypes were called using QuantStudio Real-Time PCR Software (Thermo Fisher Scientific, Waltham, MA) and compared to a cohort of 376 previously genotyped neurologically normal controls ascertained at Mayo Clinic [51].

**Screening for genetic mutations.** Amplification by PCR of exons 0-12 and the 3′ untranslated region of the *GRN* gene, as well as exons 1, 7, and 9-13 of the *MAPT* gene, was performed[36,68]. Purification of the amplicons from the PCR was undertaken[36,68,69]. To assess for the presence of an expanded GGGGCC hexanucleotide repeat in *C9ORF72*, the repeat primed PCR was also used (see Supplementary Table 5 for primers) [69].

**Tau biochemistry.** Biochemistry was performed in eight PAOS-CBD cases and 20 CBS-CBD cases that were matched by age at death, sex, and co-pathology, as well as five PAOS-PSP cases and 14 matched Richardson's syndrome cases with PSP from the CurePSP Brain Bank. Frozen tissues samples were dissected from the superior frontal gyrus (SMA) and homogenized (w/v) in 5 volumes of cold 1× TBS (50-mM Tris, 150-mM NaCl, 1-mM PMSF, 1x protease inhibitor cocktail (Thermo Fisher Waltham MA) and phosphatase inhibitor cocktails A/B (Biotool Houston TX)). These 400 μl aliquots were snap frozen on dry ice and stored at −80 °C until use. Homogenates were thawed on ice and centrifuged at $100,000 \times g$ for 60 min at 4 °C in a Beckman TLA55 rotor (Beckman Coulter Brea CA), and the soluble fractions were collected as S1 and stored as above. Pellets were suspended in 400 μl of high salt buffer (10-mM Tris-HCl, 800-mM NaCl, 10% sucrose, 1-mM ethylene glycol tetraacetic acid and 1-mM PMSF at pH 7.5) and clarified at 18,000× for 20 min at 4 °C in a TLA55 rotor. The resulting supernatant was supplemented with 1% sarkosyl for 1 h at 37 °C, and the pellet was collected at $100,000 \times g$ for 2 h at 4 °C in a TLA55. These sarkosyl-insoluble P3 pellets were resuspended in 100 μl of 1× Laemmli sample buffer (Thermo Fisher) supplemented with 10% β-mercaptoethanol. For western blotting, insoluble tau samples were separated on 10% Tris-glycine gels and transferred to nitrocellulose membranes. Blots were run in duplicates. Full-sized original blots can be found in Supplementary Fig. 10. They were probed using the rabbit primary antibody, E1, raised against exon 1 of human tau at 1:1000 (Leonard Petrucelli, Mayo Clinic, Jacksonville, FL). CBD and PSP blots of insoluble tau were reprobed with PHF1 at 1:500 (Peter Davies Albert Einstein University Bronx NY) to examine endogenous degradation products of the tau filament core [22].

**Statistical analysis.** The main goal of our analyses were to assess differences between PSP and CBD, since these are the most common pathologies underlying PAOS. We used Kruskal-Wallis tests for continuous variables and Fisher's exact tests for categorical variables to assess for differences in clinical and demographic measures at baseline. For each variable, we also calculated the area under the receiver operator curve (AUROC) and 95% CI from a univariate logistic regression to show the marginal discriminatory power of each metric, i.e., effect size, since classification was the overarching goal of these analyses. We repeated these univariate analyses after calculating annualized change scores in each continuous clinical measure using all available longitudinal data in each measure. These annualized change metrics were calculated by fitting within-person ordinary least squares models with predictors for intercept (baseline score) and annualized change, then reporting the coefficient for the annualized change term as the annualized change for that individual. This allowed us to use as much data as

possible when more than two visits were available. A total of 88 data points (clinical visits) were utilized in the analysis. In addition, we modeled the trajectory of change over time in the four main clinical domains of cognition, parkinsonism, aphasia, and AOS using mixed-effects models (Supplementary Text).

To test our hypothesis that AOS subtype and age at onset help predict the specific 4R tauopathy, logistic regression was used to build a classification model that separates PSP from CBD. To report the overall classification ability of this model, we used the predicted values to calculate an AUROC and 95% CI using a Wilcoxon rank-sum test.

We fit modality-specific, whole-brain Bayesian hierarchical models to assess whether there are cross-sectional and longitudinal differences in PSP and CBD in gray-matter volumes, FDG SUVR, and DTI FA. The models were anchored at baseline as time zero, which allowed us to assess for similarities and differences when the patient first presents as well as similarities and differences in how these diseases progress through their clinical course. Hierarchical models use the idea of partial pooling of variance parameters to increase statistical power and stabilize estimates across regions while allowing us to use multiple regions-per-scan and scans-per-person, i.e., to maximize the amount of data included in these comprehensive models[70]. Each model used a neuroimaging measure as the outcome predicted by region and diagnosis, using years-from-baseline assessment as the timescale. More details of these Bayesian hierarchical models can be found in the Supplementary Text including an algebraic representation of the model as well as technical details about prior distributions and model fits.

In addition, the following secondary neuroimaging analyses were performed (see Supplementary Text for more details): (1) we assessed whether previously published relationships between AOS and aphasia and specific brain regions differed between PAOS-CBD and PAOS-PSP. (2) In order to interrogate the corticostriatal and PNL networks across AOS subtypes identified in our pathological analysis, we performed a neuroimaging comparison of the same regions between phonetic and prosodic AOS subtypes.

All analyses were performed using the statistical software R (https://www.R-project.org/) version 3.6.2 in conjunction with the rstanarm package version 2.21.1 running STAN version 2.21.0 (https://mc-stan.org/rstanarm) and the lme4 package version 1.1–25.

**Reporting summary**. Further information on research design is available in the Nature Research Reporting Summary linked to this article.

## Data availability

The data that support the findings of this study are available from the corresponding author upon reasonable request. Source data are provided with this paper.

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

## Acknowledgements

We wish to thank Kris Johnson, Linda Rousseau, Virginia Phillips, and Monica Casey-Castanedes for pathological support. This study was funded by the National Institutes of Health, grants R01-DC12519, R01-DC010367, R01-DC14942, R01-NS89757, and UL1TR002377.

## Author contributions

All authors have made substantial contributions as outlined below, have approved the submitted version of the manuscript, and agree to be personally accountable for the authors own contributions. K.A.J. and J.L.W. had access to all data reported in the manuscript and were responsible for study concept and design, data interpretation and drafting the original report which was reviewed and revised by all co-authors. K.A.J., J.R. D., H.M.C., R.L.U., H.B., M.M.M., M.Bu., E.A.S., J.S. and F.A. were responsible for acquisition of clinical data. O.A.R., M.A.D., N.E.T., M.Ba., and R.R. were responsible for acquisition of biochemical and genetic data. D.W.D., R.R.R. and E.H.B. were responsible for pathological classification and semi-quantitative pathological data. J.L.W., M.L.S., C. G.S., R.I.R., and A.J.S. were responsible for all neuroimaging analyses. N.T.T., P.R.M., and E.J.P. were responsible for statistical analysis and for creating all figures. C.R.J. and V.J.L. were responsible for acquisition of MRI and FDG-PET scans. Funding was obtained by K.A.J., and J.L.W..

## Competing interests

C.R.J. serves on an independent data monitoring board for Roche, has consulted for and served as a speaker for Eisai, and consulted for Biogen, but he receives no personal compensation from any commercial entity. V.J.L. is a consultant for AVID Radio-pharmaceuticals, Eisai Co. Inc., Bayer Schering Pharma, GE Healthcare, and Merck Research, and receives research support from GE Healthcare, Siemens Molecular Imaging, and AVID Radiopharmaceuticals. M.L.S. owns stocks in Align Technology, Inc., Inovio Pharmaceuticals Inc., LHC Group, Inc., Mesa Laboratories, Inc., Natus Medical Inc., and Varex Imaging Corporation. All other authors declare no competing interests.
