## [Peer Review File · Nature Communications]

Reviewers' Comments:

Reviewer #1:

Remarks to the Author:

The authors report results of a longitudinal (10-year) study of patients with progressive apraxia of speech (PAOS) with or without aphasia or symptoms of corticobasal syndrome or Richardson syndrome. The patients underwent detailed and standardized clinical batteries at each visit, and multiple analytic techniques were performed to identify genetic, neuroanatomical, and pathological differences between subgroups. It is certainly the most thorough description of PAOS that has been published.

The paper is clearly written, and the information is novel. My main concern is that subdividing the patients in various ways did not reveal anything very useful in terms of prognosis, management, or even understanding the mechanism underlying the condition. The report, "The results of this study offer unique insights into the neurobiology of PAOS that have important implications for patient prognosis and future development of experimental therapeutic paradigms." However, they did not really explain these implications. Some examples of how results might guide prognosis or management would help.

Their main hypothesis was "that genetic and biochemical characteristics of PAOS-CBD and PAOS-PSP would be similar to those previously reported in CBD and PSP when associated with their more classic presentations of corticobasal syndrome (CBS) and Richardson's syndrome, respectively."

The question is whether (or why) anyone would have thought otherwise?

The main difference they found between PAOS-CBD and PAOS-PSP was a greater frequency of PAOS type 1, or phonetic type (characterized by the relative predominance of articulatory distortions, distorted sound substitutions or additions and articulatory groping) and in PAOS-CBD and greater frequency of POAS type 2, or prosodic type (predominance of slow speech rate and segmentation within multi-syllabic words or across words) in the PSP cases. The mean age also differed, but the sample size is a bit small to make much of that, especially as the age range was overlapping.

There were no other clinical features that differentiated between the 2 groups, and there was a large degree of overlap in patterns of white matter degeneration. There were also no differences in the insoluble tau subunits forming the filamentous aggregates in the PAOS-CBD and PAOS-PSP cases, leading them to conclude, "there is no biochemical evidence to suggest that PAOS-CBD and PAOS-PSP represent distinct tauopathies." There were, by definition, differences in the brain regions affected by the pathology.

The clinical course was highly heterogenous within groups, and overlapping between groups, making it very difficult to draw conclusions about prognosis. They found:

Over time, of the 17 PAOS-CBD patients, 13 evolved into a CBS, two into behavioral variant frontotemporal dementia, one into frontotemporal dementia with parkinsonism, and one into Richardson's syndrome. . Of the 10 PAOS-PSP patients, five evolved into Richardson's syndrome and three into a CBS; the final two patients did not have longitudinal follow-up. The AOS subtype progressed to become mixed in 15 of 23 patients (65%) with follow-up clinical assessments, without discernable differences in conversion between PAOS-CBD and PAOS-PSP, or between phonetic and prosodic patients.

Minor:

This sentence is hard to parse:

"It is unknown, however, whether molecular pathology differs when AOS presents as a pure entity (PPAOS), is associated with aphasia, or is embedded within another neurodegenerative syndrome." Perhaps it would be more clear if they punctuate it differently: It is unknown, however, whether molecular pathology differs when AOS: (1) presents as a pure entity (PPAOS), (2) is associated with aphasia, or (3) is embedded within another neurodegenerative syndrome.

Reviewer #2:

Remarks to the Author:

Thank you very much for the opportunity to review this interesting manuscript, which I have read with great interest.

This is a very well written paper by a strong research team that evidences both diagnostic and methodological rigor. In my opinion, this will become one of the most influential papers in neurodegenerative disorders in the following years, specifically in the fields of apraxia of speech, primary progressive aphasia, and atypical parkinsonisms.

In this study, the authors report the findings of a cohort of 32 patients with progressive apraxia of speech that underwent neuropathological assessment at death. These patients were examined with a comprehensive clinical, language/speech, multimodal neuroimaging, and genetics protocol. Main diagnostic tools were done at baseline and longitudinally, which is another strength of the study.

The authors clearly demonstrate that PAOS is closely linked to both corticobasal degeneration and progressive supranuclear pathologies. Interestingly, the authors show the relevance of distinguishing between PAOS-phonetic and PAOS-prosodic subtypes, which are predictive of the underlying pathology. This distinction was introduced recently in the literature by the same group (Utianski et al. Brain Lang 2018), and this manuscript confirms the convenience of adding this approach to improve the clinical-pathological correlation. Furthermore, all the longitudinal, neuroimaging, genetics and pathological information is novel and makes a great contribution to an accurate description of PAOS. Thus, in my opinion, this study provides a major contribution to the existing literature, and this it supports the need to update the current criteria for diagnosis and classification of the speech/language-onset neurodegenerative disorders.

I have some minor suggestions/questions:

-I am not sure whether the abstract is enough explicative, although it could be due to the length limitations. Maybe some mention to the association between PAOS-CBD and phonetic AOS, and PAOS-PSP and prosodic AOS may be appropriate if possible.

-Was AOS subtype classification performed blinded to other diagnostic tools (MRI, PET, etc.)?

-It would be interesting to know how long the patients last to develop the second syndrome (parkinsonisms, dementia). I am aware that this could be difficult to estimate because of the slow onset of symptoms in these disorders, but an estimate of the time to develop a non-speech/language deficit may be very interesting.

-How did the authors the neuroimaging longitudinal preprocessing? Were images of each patient coregistered? Please, further specify this section to improve the reproducibility of the methods.

-Table 1: regarding family history: what did the authors mean? Family history of dementia? Of any neurological disorder? Of apraxia of speech? Please clarify this point.

-Statistical analysis seems correct and appropriate for this study. I have no comments specifically about this.

-I am some bit confused about the follow-up. If I am correct, patients lasted 2.7 and 5.3 years until death (baseline to final follow-up, years). It seems a very short time to dead due to some cause related to the PAOS. This could also be interpreted as a selection bias, in which for this study, only patients with a very aggressive course were included. How the authors explain this short time? I suggest including a commentary in the Discussion, and the cause of dead. And maybe a flowchart with all the patients included in the NIH project as Supplementary Material, showing those selected for this study and the other currently under follow-up may be useful to clarify this point.

-Number of PET and MRI scans included in the analysis should be specified (e.g. median number of scans per patient).

Dr. Jordi A Matias-Guiu.

Reviewer #3:

Remarks to the Author:

Josephs and colleagues report imaging, genetic, biochemical and pathology data on a prospective cohort of AOS patients with autopsy confirmation n=32. They test the hypothesis that underlying pathological substrate would be more frequently 4R tauopathies and previously proposed subtypes of AOS by these authors would relate to specific tauopathies (CBD= phonetic, PSP=prosodic). They find these relationships exist in their cohort with neuroimaging signatures unique to each pathology as well as novel findings of a much lower frequency of H1 haplotype in PSP-AOS compared to a reference cohort of PSP without AOS. Another important finding is that AOS when in combination with other FTD syndromes (e.g. bvFTD- what they call AOS+) is more likely to be GRN + FTLT-DTP.

The strengths of this study include longitudinal follow up to autopsy with careful consensus for clinical designation of patients and multimodal imaging during life. The results are generally supported by the data for CBD and PSP tauopathies having distinct neuroimaging and clinical features of AOS and are important in the field in neurodegeneration. One weakness is that the designation of the AOS subtypes appear to be driven at least in part by many of the patients in this series and therefore relatively circular, therefore the hypotheses being tested above appear to be generated from some of the earlier reports of a subset of these patients leading the results to be more descriptive in nature. It would be helpful to acknowledge numbers of patients in this cohort who have been previously reported to help determine how much novelty are in these results and how generalizable they may be outside of this cohort.

An interesting finding that is potentially more impactful to the field of cognitive neuroscience more broadly is the pathological findings of more subcortical neuron loss in nigral-globus pallidus-subthalamic nucleus vs more neocortical neuron loss in those patients with prosodic variant of AOS vs phonemic AOS irrespective of CBD vs PSP pathology. This is difficult to measure postmortem and data presented suggests a trend but it is unclear why this contrast was not pursued in imaging, although subcortical regions similarly are difficult to study in vivo. Perhaps it is a missed opportunity that specific language features of AOS were not related directly to imaging and pathology and contrasted with pathological subtype rather than the syndromic subgroup categories of AOS proposed.

Finally, there is limited discussion or interrogation of the language network in this study, it would be very important to note how SMA and subcortical region pathology interacts with areas in language production and comprehension.

Other minor comments include:

It would be helpful to include more details on the TDP subtype and regional distribution of TDP and AGD in the AGD+TDP subtype to determine which (or if both) pathologies regionally match distribution of language production to be likely substrate of AOS in that case.

The clinical diagnosis of "frontotemporal dementia with parkinsonism" is mentioned for one case but this term lacks clinical criteria and classically linked with MAPT mutations, it would be helpful to clarify.

It would be helpful to note if any of these cases had features of globular glial tauopathy which Josephs, Dickson and colleagues have helped pioneer as a novel 4R tauopathy.

REVIEWER COMMENTS

Reviewer #1 (Remarks to the Author):

The authors report results of a longitudinal (10-year) study of patients with progressive apraxia of speech (PAOS) with or without aphasia or symptoms of corticobasal syndrome or Richardson syndrome. The patients underwent detailed and standardized clinical batteries at each visit, and multiple analytic techniques were performed to identify genetic, neuroanatomical, and pathological differences between subgroups. It is certainly the most thorough description of PAOS that has been published. The paper is clearly written, and the information is novel.

My main concern is that subdividing the patients in various ways did not reveal anything very useful in terms of prognosis, management, or even understanding the mechanism underlying the condition. The report, "The results of this study offer unique insights into the neurobiology of PAOS that have important implications for patient prognosis and future development of experimental therapeutic paradigms." However, they did not really explain these implications. Some examples of how results might guide prognosis or management would help.

RESPONSE: The main purpose of this paper was to understand the neurobiology of PAOS. The study included patients in which AOS was the sole presenting feature, i.e. PPAOS, those with mixed AOS and aphasia and those in which AOS accompanies other neurological features (+AOS). The results of this study provide genetic, biochemical, clinical and pathological detail about PAOS, a devastating disorder that renders patients mute, often wheelchair bound and shortens their lifespan. No such study exists in the literature and, therefore, we feel that all of our results have value to the scientific community, to physicians who will encounter such patients, and to patients and families, even if some results are negative. The pathology of PAOS was predominantly that of either CBD or PSP, and, hence, we sought to determine whether there were differences between those that died and had one or the other pathology.

AOS subtype is a characteristic of PAOS and this study helps to validate the distinction of the two subtypes by showing that they have different underlying pathologies and different network characteristics. We also found that AOS subtype is important in terms of understanding the evolution of PAOS. The study shows that patients with the phonetic subtype are more likely to develop the corticobasal syndrome, while those with the prosodic subtype are more likely to develop Richardson's syndrome. These two syndromes affect different aspects of motor function and although to the reviewer it may not seem important to differentiate them, most patients and families appreciate a discussion of differences which helps them to make decisions about their future. For example, explaining to a patient who has isolated problems with their speech that over time he or she will begin to fall and likely lose the ability to walk unassisted, versus not being able to use the right arm, may not seem important but it is. This is the discussion we have with every single PAOS patient when the carer asks what is the prognosis? In addition, physical therapy will differ in patients with balance issues and falls compared to those with asymmetric limb apraxia. Hence, the results have substantial implications for patient and family counseling about the disease course.

The clinical findings regarding the distinction between PAOS with and without aphasia, and the distinction between phonetic and prosodic subtypes also have implications for speech-language therapy/management approaches. Thus, for example, people with PAOS without aphasia have a slower course/longer survival and better prospects for maintaining language skills sufficient for day-

to-day communication and handling work demands than do patients who are also aphasic; patients with the phonetic subtype have faster rates of decline in language ability and may be at greater risk for the emergence of behavioral and other cognitive problems. This has enormous value in the early-mid stage of the disease when counseling patients and their significant others about the degree to which the disorder can be expected to change over time and, for example, long term planning regarding employment and when and what form alternative means of communication may eventually be required. The specific speech therapy approaches to maintaining or even temporarily improving communication skills are quite different for PAOS patients with aphasia. Patients with the prosodic subtype generally will progress more slowly relative to needs for alternative communication and alternative communication strategies may be more effective and easier to implement in the absence of aphasia.

Regarding “mechanism underlying the condition” the different clinical and neuroimaging findings suggest that the locus within the speech-language network, and the migration of disease through it, differs between the conditions, with the prosodic subtype being more tightly linked to “motor” networks and the phonetic subtype relatively more tightly linked to “cortical” networks; at the least, this is of theoretical importance. In addition, the significant age of onset differences between the phonetic vs prosodic subtypes also raise the possibility of an interaction between pathophysiology and age-related vulnerability in different components of the motor speech planning/programming network (Utianski, Duffy et al., 2018).

Regarding the implications for the future development of experimental therapeutic paradigms, our network-level findings could be utilized in future therapies that could be administered to focally target one part of the network. This could pertain to differences in regional targeting or neurochemical targeting since the neurotransmitters in the cortical network are different from those in the motor network. It could also have implications on the development of treatments that target molecular features of CBD versus those of PSP. Our findings would also have implications for the design of clinical treatment trials. Our results show that clinical trials that target patients suspected to have underlying PSP or CBD would significantly benefit from enrolling prosodic and phonetic phenotypes, respectively, given that these represent the mildest clinical presentations of these underlying pathologies. The different network and imaging features also suggest that different neuroimaging biomarkers may be needed to adequately track disease progression and treatment affects in phonetic and prosodic subtypes of AOS.

We have now added an analysis assessing trajectories of change in general cognition, motor parkinsonism, global aphasia and AOS severity over time from disease onset (Supplementary Figure 2). This analysis helps to support the prognostic implications of this study which we also now discuss in more detail on page 20.

Their main hypothesis was “that genetic and biochemical characteristics of PAOS-CBD and PAOS-PSP would be similar to those previously reported in CBD and PSP when associated with their more classic presentations of corticobasal syndrome (CBS) and Richardson’s syndrome, respectively.” The question is whether (or why) anyone would have thought otherwise?

RESPONSE: There are many reasons why one would want to investigate genetics and biochemistry in this cohort of PAOS patients, and why one could have imagined they may differ from classic PSP and CBD. The tauopathies are an extremely complicated set of neurodegenerative diseases that are heterogeneous in terms of clinical presentation, molecular pathology and biochemistry. In fact, a

study just published in *Brain Pathology* in March 2021 (Llibre-Guerra et al. *Brain Pathology*, March 2021) identified a novel tauopathy associated with a *TMEM106B* gene polymorphism. This polymorphism has been previously linked to a completely different protein called TDP-43 and hence one could argue there was no reason to look for an association with tau, i.e. why would anyone have thought this would be the case? Given this very recent finding, we added a new genetic analysis looking for an association between this *TMEM106B* gene polymorphism and our PAOS patients. The patients in our cohort had PSP and CBD pathology which are both 4R tauopathies, and although one may argue that there is no difference between PSP and CBD since both show the upper doublet bands at 64 and 68 kDa, it has been known for almost 20 years that there are in fact biochemical differences in the low molecular weight tau fragments in both diseases, with PSP showing a band of 33kDa and CBD a band of 37kDa. These bands have been shown to consist of the carboxyl terminal fragments of tau but with different amino termini. Hence, although the isoforms of tau may be similar across 4R tauopathies it is clear that there can be different proteolytical processing of tau which helps us to better understand these diseases. Tau deposition may differ in terms of C and N terminal fragments and post-translational modifications of the protein. In fact, the research community has failed to identify any treatment for the 4R tauopathies, likely because we do not truly understand its biochemistry. Given that we identified many unique features of the pathologies in our PAOS patients we felt it was important for us to determine whether the insoluble tau deposited in our patients was any different from what is typically seen in CBD and PSP. One of our patients, for example, with PSP pathology showed corticospinal tract degeneration in which we identified tau immunoreactive inclusions that were Gallyas positive with absence of astrocytic pathology. In this patient, as we showed in Supplementary Fig. 1, we identified globose neurofibrillary tangles in some Betz cells, i.e. in motor neurons, in laminar VI of the cortex. Surprisingly the tangle could also be seen with thioflavin S. PSP globose neurofibrillary tangles are negative on thioflavin S. Hence this finding is novel, not being previously reported in the literature. Another one of our patients with a diagnosis of CBD showed the classic features of CBD, i.e. astrocytic plaques in cardinal regions of the brain. However, unlike in classic CBD, there was striking degeneration of the substantia nigra, globus pallidus and subthalamic nucleus. This pathological finding of CBD with prominent pallidonigrolyusian degeneration is novel and also never before described in the literature. For these reasons and others, we felt it was important to assess biochemistry and genetics of our PAOS patients. Furthermore, regarding genetics, the frequency and associations of the APOE e4 allele have been found to differ among young and old-onset Alzheimer's disease, and between typical and atypical clinical variants of Alzheimer's disease. Hence, there is every reason to believe that genetics may have different associations amongst different clinical variants of PSP and CBD, particularly when there are age differences.

The main difference they found between PAOS-CBD and PAOS-PSP was a greater frequency of PAOS type 1, or phonetic type (characterized by the relative predominance of articulatory distortions, distorted sound substitutions or additions and articulatory groping) and in PAOS-CBD and greater frequency of POAS type 2, or prosodic type (predominance of slow speech rate and segmentation within multi-syllabic words or across words) in the PSP cases. The mean age also differed, but the sample size is a bit small to make much of that, especially as the age range was overlapping.

RESPONSE: We respectfully disagree with the reviewer that the sample size is too small to make much of the age difference. The ages at onset and at baseline did overlap, but not by much, and the AUROCs in Table 1 were high and the p-values very low suggesting a striking difference between groups. In fact, the p-values would survive almost any post-hoc familywise error p-value correction (FDR or Bonferroni) and thus we are confident it is a "real" result.

There were no other clinical features that differentiated between the 2 groups, and there was a large degree of overlap in patterns of white matter degeneration. There were also no differences in the insoluble tau subunits forming the filamentous aggregates in the PAOS-CBD and PAOS-PSP cases, leading them to conclude, “there is no biochemical evidence to suggest that PAOS-CBD and PAOS-PSP represent distinct tauopathies.” There were, by definition, differences in the brain regions affected by the pathology. The clinical course was highly heterogeneous within groups, and overlapping between groups, making it very difficult to draw conclusions about prognosis. They found:

Over time, of the 17 PAOS-CBD patients, 13 evolved into a CBS, two into behavioral variant frontotemporal dementia, one into frontotemporal dementia with parkinsonism, and one into Richardson’s syndrome. . Of the 10 PAOS-PSP patients, five evolved into Richardson’s syndrome and three into a CBS; the final two patients did not have longitudinal follow-up. The AOS subtype progressed to become mixed in 15 of 23 patients (65%) with follow-up clinical assessments, without discernable differences in conversion between PAOS-CBD and PAOS-PSP, or between phonetic and prosodic patients.

RESPONSE: We appreciate the reviewer summarizing the findings. As we stress above, we believe the clinical differences we identified are meaningful and important for patient prognosis. We have also now added clinical trajectory plots, as stated above, to further demonstrate differences between PAOS-CBD and PAOS-PSP which were striking for the development of aphasia. This has significant consequences. A patient with AOS can still communicate using devices that require intact language. On the other hand, a patient who has AOS and aphasia is unable to use such devices to help communication and, therefore, the prognosis is very different.

Regarding the biochemistry, the finding that PAOS-CBD and PAOS-PSP do not differ from the typical clinical manifestations of CBD and PSP respectively does not mean that the finding is not important or that the PAOS 4R tauopathies are not unique diseases. In fact, in 2006, Josephs et al described a disease called PSP with corticospinal tract degeneration. Some of these patients had a speech and language problem. Recently, in 2020, Kametani et al. (Frontiers in Neuroscience) showed that the biochemistry of this disease and PSP are similar. Yet, PSP with corticospinal tract degeneration is classified as a distinct class of tauopathy known as globular glial tauopathy. Hence, although the biochemistry was the same, morphological characteristics supported a unique classification that may ultimately lead to the identification of novel associated genes.

Minor:

This sentence is hard to parse:

“It is unknown, however, whether molecular pathology differs when AOS presents as a pure entity (PPAOS), is associated with aphasia, or is embedded within another neurodegenerative syndrome.” Perhaps it would be more clear if they punctuate it differently: It is unknown, however, whether molecular pathology differs when AOS: (1) presents as a pure entity (PPAOS), (2) is associated with aphasia, or (3) is embedded within another neurodegenerative syndrome.

RESPONSE: We have made the changes as suggested to the sentence on page 5.

Reviewer #2 (Remarks to the Author):

Thank you very much for the opportunity to review this interesting manuscript, which I have read with great interest. This is a very well written paper by a strong research team that evidences both diagnostic and methodological rigor. In my opinion, this will become one of the most influential papers in

neurodegenerative disorders in the following years, specifically in the fields of apraxia of speech, primary progressive aphasia, and atypical parkinsonism. In this study, the authors report the findings of a cohort of 32 patients with progressive apraxia of speech that underwent neuropathological assessment at death. These patients were examined with a comprehensive clinical, language/speech, multimodal neuroimaging, and genetics protocol. Main diagnostic tools were done at baseline and longitudinally, which is another strength of the study.

The authors clearly demonstrate that PAOS is closely linked to both corticobasal degeneration and progressive supranuclear pathologies. Interestingly, the authors show the relevance of distinguishing between PAOS-phonetic and PAOS-prosodic subtypes, which are predictive of the underlying pathology. This distinction was introduced recently in the literature by the same group (Utianski et al. Brain Lang 2018), and this manuscript confirms the convenience of adding this approach to improve the clinical-pathological correlation. Furthermore, all the longitudinal, neuroimaging, genetics and pathological information is novel and makes a great contribution to an accurate description of PAOS. Thus, in my opinion, this study provides a major contribution to the existing literature, and it supports the need to update the current criteria for diagnosis and classification of the speech/language-onset neurodegenerative disorders.

I have some minor suggestions/questions:

-I am not sure whether the abstract is enough explicative, although it could be due to the length limitations. Maybe some mention to the association between PAOS-CBD and phonetic AOS, and PAOS-PSP and prosodic AOS may be appropriate if possible.

RESPONSE: We thank the reviewer for this comment. We have now added more information to the abstract concerning the phonetic and prosodic associations as suggested.

-Was AOS subtype classification performed blinded to other diagnostic tools (MRI, PET, etc.)?

RESPONSE: Yes, AOS subtype classification was blinded to all other diagnostic tools, including neurological, neuropsychological, and imaging findings. Speech and language diagnoses and AOS type were determined by consensus, based only on performance on the speech/language battery. We have added this to the text on page 23.

-It would be interesting to know how long the patients last to develop the second syndrome (parkinsonism, dementia). I am aware that this could be difficult to estimate because of the slow onset of symptoms in these disorders, but an estimate of the time to develop a non-speech/language deficit may be very interesting.

RESPONSE: In response to the reviewers request, we have now provided analyses and plots in Figure 2 and Supplementary Table 3 showing longitudinal decline in the MoCA (a measure of global cognitive performance) and longitudinal increase in the MDS-UPDRS III (a measure of motor parkinsonism) across the patients to illustrate the progression of cognitive impairment and parkinsonism in our cohort respectively. It is difficult to determine specific cut-points for when patients “develop” parkinsonism or cognitive impairment as we generally see a steady decline in function over time as depicted in the graphs. However, we do see evidence for an acceleration in the rate of MDS-UPDRS III increase over time after approximately five years in our patients. This acceleration appears to occur earlier in PAOS-CBD than in PAOS-PSP, although was not significant. For MoCA, we also see evidence

that cognitive decline is faster in PAOS-CBD. We now report these findings on page 9 and discuss them on page 17. We have also added more background on the development of parkinsonism in these patients to the introduction.

-How did the authors do the neuroimaging longitudinal preprocessing? Were images of each patient co-registered? Please, further specify this section to improve the reproducibility of the methods.

RESPONSE: For our study, regional data were calculated for each time-point for each patient separately using the procedures described in the methods, and then modelling was performed using longitudinal statistical methods. We did not utilize a within-subject design for the analyses. It has been shown that longitudinal pipelines for PET that involve using single-subject templates do not improve measurement precision (Schwarz et al. J Alzheimer's Dis. 2019; 67(1): 181–195). In addition, although longitudinal pipelines utilizing within-subject designs may have reduced some variability in the MRI data, we do not have an established pipeline for such methods for DTI. In response to the reviewers comment we have edited the neuroimaging section to clarify this point on pages 26 and 27.

-Table 1: regarding family history: what did the authors mean? Family history of dementia? Of any neurological disorder? Of apraxia of speech? Please clarify this point.

RESPONSE: We apologize for not being clearer. In this study, positive family history was defined as history of any neurodegenerative disease (primary progressive aphasia, frontotemporal dementia, progressive apraxia of speech, progressive supranuclear palsy, corticobasal syndrome, Alzheimer's disease <age 65, amyotrophic lateral sclerosis, Parkinson's disease, dementia with Lewy bodies) in a first- or second-degree relative. We have now added this explanation of positive family history to the text on page 20.

-Statistical analysis seems correct and appropriate for this study. I have no comments specifically about this.

-I am some bit confused about the follow-up. If I am correct, patients lasted 2.7 and 5.3 years until death (baseline to final follow-up, years). It seems a very short time to death due to some cause related to the PAOS. This could also be interpreted as a selection bias, in which for this study, only patients with a very aggressive course were included. How does the authors explain this short time? I suggest including a commentary in the Discussion, and the causes of death. And maybe a flowchart with all the patients included in the NIH project as Supplementary Material, showing those selected for this study and the other currently under follow-up may be useful to clarify this point.

RESPONSE: Please allow us to clarify. The numbers 2.7 and 5.3 (baseline to final follow-up) are not the time from disease onset to death but the time between when the first evaluation occurred and the last evaluation. We have clarified this in the table. The total disease duration (please see 3 rows above in the table) was 9 and 11 years respectively. As requested, we have added a swim plot (supplementary figure 8) that shows the length of time we have been following all patients recruited into the grant including the 32 patients that had an autopsy and are included in this study. As the reviewer can see, there is no obvious evidence for a selection bias.

-Number of PET and MRI scans included in the analysis should be specified (e.g. median number of scans per patient).

RESPONSE: We have now specified the median number of PET and MRI scans utilized in the analysis in Table 1.

Reviewer #3 (Remarks to the Author):

Josephs and colleagues report imaging, genetic, biochemical and pathology data on a prospective cohort of AOS patients with autopsy confirmation n=32. They test the hypothesis that underlying pathological substrate would be more frequently 4R tauopathies and previously proposed subtypes of AOS by these authors would relate to specific tauopathies (CBD= phonetic, PSP=prosodic). They find these relationships exist in their cohort with neuroimaging signatures unique to each pathology as well as novel findings of a much lower frequency of H1 haplotype in PSP-AOS compared to a reference cohort of PSP without AOS. Another important finding is that AOS when in combination with other FTD syndromes (e.g. bvFTD- what they call AOS+) is more likely to be GRN + FTLN-TDP.

The strengths of this study include longitudinal follow up to autopsy with careful consensus for clinical designation of patients and multimodal imaging during life. The results are generally supported by the data for CBD and PSP tauopathies having distinct neuroimaging and clinical features of AOS and are important in the field in neurodegeneration. One weakness is that the designation of the AOS subtypes appear to be driven at least in part by many of the patients in this series and therefore relatively circular, therefore the hypotheses being tested above appear to be generated from some of the earlier reports of a subset of these patients leading the results to be more descriptive in nature. It would be helpful to acknowledge numbers of patients in this cohort who have been previously reported to help determine how much novelty are in these results and how generalizable they may be outside of this cohort.

RESPONSE: We thank the reviewer for carefully reading our manuscript and summarizing the findings. We apologize if we were not clearer regarding the AOS subtype-pathological relationship which may be why the reviewer made the comment about circularity which is not correct. To clarify, we first described AOS subtypes in 2013 (Josephs et al. Neurology) and have published another paper changing the names from subtype 1 and subtype 2 to phonetic and prosodic subtype. However, we have never assessed the relationship between AOS subtype and pathology, even in any subject. In fact, we did not know the pathology of AOS subtype until we began to analyze the data for this manuscript. In addition, all AOS subtypes are performed during life (beginning in 2010) prior to autopsy and we do not change the clinical diagnosis of AOS subtype once rendered. In other words, we clinically diagnose the patients with an AOS subtype once he/she is enrolled into the R01 and we then follow them to death and determine pathology at the time of death. Therefore, AOS subtyping is 100% independent of pathological diagnosis. We hope this explanation of the study methods help to clarify that there is no circularity. Furthermore, only 10 of the patients in this study were in our original 2013 paper describing AOS subtypes. The other 22 were not. We have added this to the manuscript on page 22.

An interesting finding that is potentially more impactful to the field of cognitive neuroscience more broadly is the pathological findings of more subcortical neuron loss in nigral-globus pallidus-subthalamic nucleus vs more neocortical neuron loss in those patients with prosodic variant of AOS vs phonemic AOS irrespective of CBD vs PSP pathology. This is difficult to measure postmortem and data presented suggests a trend, but it is unclear why this contrast was not pursued in imaging, although subcortical regions similarly are difficult to study in vivo.

RESPONSE: We appreciate the reviewer's suggestion to pursue our pathological findings on neuroimaging. We had not pursued this angle before as we were worried about segmentation and measuring small subcortical regions in vivo. However, per the reviewer's request we proceeded and now add subthalamic nucleus, globus pallidus and substantia nigra volumes to all our imaging analyses. Interestingly, as the reviewer can see (new Supplementary Figure 6), we did find evidence for greater volume loss in the PNL nuclei in patients with the prosodic subtype, as we did on pathology. We also saw trends for greater corticostriatal involvement in the phonetic patients, as with pathology. We are thankful for this comment as we think this added imaging analysis is a nice addition/validation to our study. We report the results of this analysis on page 13 and discuss the findings on page 17.

Perhaps it is a missed opportunity that specific language features of AOS were not related directly to imaging and pathology and contrasted with pathological subtype rather than the syndromic subgroup categories of AOS proposed. Finally, there is limited discussion or interrogation of the language network in this study, it would be very important to note how SMA and subcortical region pathology interacts with areas in language production and comprehension.

RESPONSE: The primary language feature of PAOS is agrammatic aphasia. We must admit that we did not focus on aphasia in this study and instead focused on AOS. With-that-said, we did include Broca's area in our original submission albeit referred to it as frontal inferior; we now refer to it as Broca's area, to be more specific. Our goal is still to maintain focus on AOS in this paper. However, given the reviewers comment we have added more analyses specific to language. We have performed correlation analyses between two different measures of language (the Token test a measure of spoken language comprehension and the Western Aphasia Battery Aphasia Quotient a measure of global aphasia severity) and two regions that have previously been shown to correlate with language abnormalities (Broca's area and left superior temporal gyrus), contrasted by pathology subtype (CBD vs PSP). We also performed a similar analysis for AOS severity measured by the ASRS, assessing regions that have previously been associated with AOS (SMA and precentral gyrus). We performed these analyses using FDG-PET SUVR data. Please see New Supplementary Figure 5 for results, as well as additions to the text on page 12 and 17. We also performed a semi-quantitative pathological analysis of neuronal loss and tau burden for additional language areas, i.e. the superior temporal lobe and inferior parietal (angular and supramarginal gyri). However, the results lacked adequate range to allow us to correlate these findings with clinical measures of aphasia, with almost all patients showing no neuronal loss in these regions. We were unable to perform histological analyses on Broca's area because we do not have available tissue on this area for most cases.

Other minor comments include:

It would be helpful to include more details on the TDP subtype and regional distribution of TDP and AGD in the AGD+TDP subtype to determine which (or if both) pathologies regionally match distribution of language production to be likely substrate of AOS in that case.

RESPONSE: We have now performed semi-quantitation of both TDP-43 and AGD lesion count for the following regions: Superior Frontal, Peri-Rolandic, Superior Temporal, Inferior Parietal, Cingulate Basal nucleus of Meynert, Amygdala, Entorhinal cortex, Hippocampus CA1, Hippocampus dentate, Basal ganglia, Midbrain Tegmentum, Medullary Tegmentum, Medullary CN XII, and Medullary Inferior Olive. Given that there was no AGD beyond limbic cortex, one could speculate that it was the TDP-43 causing the AOS. We show this data in new Supplementary Table 4. In addition, we also now state that the TDP-43 subtype was type B. We appreciate this comment as the findings were very helpful,

specifically the issue of this patient having a type of motor neuron disease was considered while she was alive but EMG and other tests could not identify evidence of motor neuron disease. After performing the semi-quantitative analysis, it is now obvious that there were subtle features of motor neuron disease pathologically, such as rare inclusions within cranial nerve XII in the absence of anterior horn cell loss or Bunina bodies. This fits with the short disease duration and the subtype of FTLD-TDP type B. We now refer to this patient as FTLD-TDP B + AGD, rather than AGD + TDP.

The clinical diagnosis of “frontotemporal dementia with parkinsonism” is mentioned for one case but this term lacks clinical criteria and classically linked with MAPT mutations, it would be helpful to clarify.

RESPONSE: We have tried to clarify what we mean on page 9. The patient had mixed features of bvFTD and moderate levodopa-unresponsive parkinsonism. We agree that there is no formal clinical diagnosis for this syndrome and hence we are being descriptive.

It would be helpful to note if any of these cases had features of globular glial tauopathy which Josephs, Dickson and colleagues have helped pioneer as a novel 4R tauopathy.

RESPONSE: None of the cases met criteria for a globular glial tauopathy (GGT). One atypical PSP case did have evidence of corticospinal tract degeneration with a globose neurofibrillary tangle observed in a Betz cell in the motor cortex. However, there were no globular oligodendroglial or astrocytic inclusions and Gallyas silver stain did label the astrocytic lesions which were predominantly coiled bodies. We have clarified this point on page 7.

Reviewers' Comments:

Reviewer #1:

Remarks to the Author:

The authors have thoughtfully responded to all of my concerns. I do not think the paper can or needs to be further improved. It is a very comprehensive study of an interesting topic.

Reviewer #2:

Remarks to the Author:

The authors have addressed all my comments and suggestions.

Reviewer #3:

Remarks to the Author:

The authors have been extremely responsive to previous round of comments including adding details of previously studied patients in this cohort and new imaging , genetic and longitudinal clinical analyses. The findings of distinct features of AOS between PSP and CBD tauopathies and conversely shared anatomic features of AOS subtypes independent of pathology is interesting as well as the novel association of H2 with PSP/CBD with AOS compared to other clinical phenotypes. The only minor remaining comments are to consider adding the n=10 of previously reported patients to the manuscript as even though pathology was not previously reported in these cases the criteria for AOS subtypes was derived in part from this subgroup of the total cohort. It would also be helpful to include the specific locus on TMEM106B genotyping was performed.

Reviewer #3 (Remarks to the Author):

The authors have been extremely responsive to previous round of comments including adding details of previously studied patients in this cohort and new imaging, genetic and longitudinal clinical analyses. The findings of distinct features of AOS between PSP and CBD tauopathies and conversely shared anatomic features of AOS subtypes independent of pathology is interesting as well as the novel association of H2 with PSP/CBD with AOS compared to other clinical phenotypes. The only minor remaining comments are to consider adding the n=10 of previously reported patients to the manuscript as even though pathology was not previously reported in these cases the criteria for AOS subtypes was derived in part from this subgroup of the total cohort.

RESPONSE: The n=10 previously reported patients are already included in the study. In response to the reviewer's comment we have highlighted those 10 original clinical patients in the Swim plot in Supplementary Figure 8 so that the readers can tell which 10 patients were in our clinical paper where we described AOS type.

It would also be helpful to include the specific locus on TMEM106B genotyping was performed.

RESPONSE: We have now included more details on the specific locus in the TMEM106B gene on which the genotyping was performed.